



# Comparison of optical-equivalent snow grain size estimates under Arctic low Sun conditions during PAMARCMiP 2018

Evelyn Jäkel[1], Tim Carlsen[1,2], André Ehrlich[1], Manfred Wendisch[1], Michael Schäfer[1], Sophie Rosenburg[1], Konstantina Nakoudi[3], Marco Zanatta[3,4], Gerit Birnbaum[5], Veit Helm[5], Andreas Herber[5], Larysa Istomina[5,6], Linlu Mei[6], and Anika Rohde[7]

[1]Leipzig Institute for Meteorology (LIM), University of Leipzig, Germany
[2]Department of Geosciences, University of Oslo, Oslo, Norway
[3]Alfred Wegener Institute Helmholtz Centre for Polar and Marine Research (AWI), Potsdam, Germany
[4]LISA (Laboratoire Interuniversitaire des Systèmes Atmosphériques), UMR CNRS 7583, Université Paris-Est-Créteil, Université de Paris, Institut Pierre Simon Laplace (IPSL), Créteil, France
[5]Alfred Wegener Institute Helmholtz Centre for Polar and Marine Research (AWI), Bremerhaven, Germany
[6]Institute of Environmental Physics, University of Bremen, Bremen, Germany
[7]Institute of Meteorology and Climate Research, Karlsruhe Institute of Technology (KIT), Karlsruhe, Germany

**Correspondence:** Evelyn Jäkel (evi.jaekel@uni-leipzig.de)

**Abstract.** The size and shape of snow grains directly impacts the reflection by a snowpack. In this article, different approaches to retrieve the optical-equivalent snow grain size ($r_{\mathrm{opt}}$) or, alternatively, the specific surface area (SSA) using satellite, airborne, and ground-based observations are compared and used to evaluate ICON-ART (ICOsahedral Non-hydrostatic – Aerosols and Reactive Trace gases) simulations. The study is focused on low Sun and partly rough surface conditions encountered during a

three-week campaign conducted North of Greenland in March/April 2018 within the framework of the PAMARCMiP (Polar Airborne Measurements and Arctic Regional Climate Model Simulation Project) project. Further, we propose an adjusted airborne retrieval method by using the albedo at 1700 nm wavelength. This reduced the effect of atmospheric masking and improved the sensitivity on $r_{\mathrm{opt}}$. From this approach we achieved a significantly improved uncertainty ($< 25\,\%$) compared to the previous method. Ground-based in situ measurements indicated an increase of $r_{\mathrm{opt}}$ of 15 µm within a five-day period after

a snowfall event, which is small compared to previous observations under similar temperature regimes. This low growth rate is well represented by a parametrization of snow metamorphism, when the vertical temperature gradient effect is suppressed. ICON-ART captured the observed change of $r_{\mathrm{opt}}$ during snow fall events, but systematically overestimated the snow grain growth by about 100 %. Satellite-based and airborne retrieval methods showed higher and more variable $r_{\mathrm{opt}}$-values over sea ice ($< 300$ µm) than over land surfaces ($< 100$ µm), which could partly be attributed to the impact of surface roughness

on the retrieval. Moderate Resolution Imaging Spectroradiometer (MODIS) retrievals revealed a large spread within a series of subsequent individual overpasses, indicating their limitations in observing the snow grain size evolution in early spring conditions with low Sun.



# 1 Introduction

The enhanced sensitivity of the Arctic climate system with regard to global warming, referred to as Arctic Amplification, is associated with a number of feedback mechanisms (Serreze and Barry, 2011; Cohen et al., 2014; Wendisch et al., 2019). Numerous scientific efforts are targeted on the quantification of individual contributions of these feedback mechanisms to

Arctic Amplification (Pithan and Mauritsen, 2014; Qu and Hall, 2014; Fletcher et al., 2015; Goosse et al., 2018; Block et al., 2020). Simulations of various climate models from the Coupled Model Intercomparison Project Phase 5 (CMIP5; Taylor et al., 2012) identified the snow–ice surface albedo feedback, driven by the positive coupling between increasing near-surface temperature and the melting of snow and sea ice, as the second largest contributor after the lapse rate feedback (Pithan and Mauritsen, 2014; Block et al., 2020). Qu and Hall (2014) noted a bias in the magnitude of the snow-ice albedo feedback, which

was attributed to the inadequate parametrization of the snow and ice surface albedo in several models.

Most climate models assume a constant surface albedo for fresh and old snow with simplistic assumptions for the transition between both extremes. In reality, the parameters determining the albedo of snow-covered surfaces are manifold, such as the solar zenith angle, cloudiness, snow impurities, surface roughness, snow grain size and shape (Choudhury and Chang, 1981; Warren, 1982; Warren et al., 1998; Dumont et al., 2010; Gardner and Sharp, 2010; Pirazzini et al., 2015; Saito et al., 2019;

Tanikawa et al., 2020; Larue et al., 2020). E.g., Donth et al. (2020) quantified the effect of black carbon (BC) impurities, cloudiness, and snow grain size on the broadband snow surface albedo ($\alpha_{bb}$). They identified a minor BC effect ($\Delta\alpha_{bb} = 0.01$ for a reasonable range of BC mass concentration in snow), but major effects due to cloudiness ($\Delta\alpha_{bb}$ up to 0.12 for aged snow) and snow grain size ($\Delta\alpha_{bb} = 0.07$ for particles of fresh and aged snow). Pirazzini (2004) observed temporal variations of the snow surface albedo which showed an even wider range between 0.58 and 0.82 in Antarctica.

Therefore, an improved understanding of the snow and ice metamorphism in terms of temporal changes in snow particle shape and size is needed. Driven by thermodynamics, snow grains constantly transform (snow metamorphism). The rate of these transformations depends on the ambient conditions, such as temperature and humidity. The snow structure changes more rapidly for higher temperatures and for greater temperature differences within the snowpack. For winter and early spring in Polar areas, the snow metamorphism runs mostly under dry conditions. Local temperature gradients in a snowpack lead to

diffusion of water vapour from higher to lower temperature areas. This diffusion is linked to sublimation of warmer grains and vapour depositional growth of colder ones, typically resulting in faceted and depth hoar grains (Colbeck, 1983; Gubler, 1985). For snowpacks with a low vertical temperature gradient (below $0.1$ K cm$^{-1}$; Taillandier et al., 2007), the so-called equilibrium or curvature growth becomes dominant (Flanner and Zender, 2006). Microscopic differences in water vapour pressure at saturation due to variable curvatures for a single snow grain cause water vapour diffusion from convex to concave

parts of the snow grain. The corresponding deposition (at concave surfaces) and sublimation (at convex surfaces) changes the particle shape to more rounded particles, which have a larger grain size than initially (Colbeck, 1982; Cabanes et al., 2002).

Due to the complex and versatile shapes of snow grains (Kikuchi et al., 2013), the snow particle size refers to an optical-equivalent grain size, rather than a geometrical measure. The optical-equivalent grain size represents a collection of spheres with the same volume-to-surface ratio compared to non-spherical snow particles (Grenfell and Warren, 1999; Neshyba et al.,





2003). As an alternative measure, the specific surface area (SSA, in units of $m^2\,kg^{-1}$) is used, which can be related to the optical-equivalent snow grain size radius ($r_{opt}$):

$$\mathrm{SSA} = \frac{3}{\rho_{ice} \cdot r_{opt}} \quad , \tag{1}$$

with $\rho_{ice}$ representing the density of ice ($917\,kg\,m^{-3}$). For simplification, in the following we use the term snow grain size,
which refers to the more accurate term optical-equivalent snow grain size.

The snow metamorphism also effects the surface radiative energy budget. More spherical and larger snow grains amplify the solar absorption and lead to an increase of the surface temperature that in turn accelerates the snow metamorphism. Larger grains allow for a deeper penetration of the incident radiation into the snowpack linked to a higher probability of absorption in the shortwave-infrared (SWIR) spectral range and a decrease of the snow surface albedo (e.g., Warren, 1982; Picard et al.,
2009). Several studies followed the evolution of the SSA by ground-based (Libois et al., 2015; Picard et al., 2016; Carlsen et al., 2017; Dumont et al., 2017), airborne (Carlsen et al., 2017), and satellite-based (Jin et al., 2008; Lyapustin et al., 2009) observations. The common basis of these studies is the retrieval of the snow grain size, which is obtained by applying the analytical SSA–snow reflection relationship derived from the radiative transfer approach by Kokhanovsky and Zege (2004). These long–term observations were also compared to snowpack models such as Crocus (Vionnet et al., 2012) to investigate
the interannual variability of the SSA decrease in summer with respect to the amount of precipitation and to improve the parametrizations of a snowpack model (Libois et al., 2015).

All common retrieval methods rely on the same asymptotic radiative transfer (ART) approach (Kokhanovsky and Zege, 2004) assuming a plane surface. The ground-based methods mostly refer to the measurements of the surface albedo, whereas satellite data provide the bidirectional reflectance distribution function (BRDF). However, both reflection properties are influenced by
(i) the macroscopic surface roughness, and (ii) the snow grain shape. Therefore, a comparison of grain size data derived from different observation platforms is significantly affected by the assumption and uncertainty of these two parameters. Previous systematical studies of roughness effects were mostly linked to sastrugi (Warren et al., 1998; Kuchiki et al., 2011) or well-orientated artificial roughness features (Larue et al., 2020). In general, an increasing surface roughness tends to reduce the surface albedo and leads to a positive bias of the retrieved snow grain size (Warren, 1982; Picard et al., 2016). The insolation-
weighted average incidence angle, which is relative to a plane snow surface, is decreased. Together with multiple reflections in the concavities, this leads to an enhanced probability of absorption (Warren et al., 1998). The amplitude of the surface albedo reduction increases for higher solar zenith angles (SZAs, Leroux and Fily, 1998), and reveals a larger effect for SWIR than for visible wavelengths due to different contributions from multiple scattering (Kuchiki et al., 2011; Larue et al., 2020). Exemplarily, Larue et al. (2020) found an albedo reduction of 0.03 − 0.04 at 700 nm wavelength and 0.06 − 0.10 at 1000
nm, depending on the fraction of roughness (13 − 63 %). Since the roughness effect becomes evident when the height of the roughness features is larger than the penetration depth of the incident radiation, smaller irregularities can reduce the SWIR albedo more effectively than the VIS albedo (Warren, 1982). For satellite-based remote sensing of the snow grain size, the deviation of the snow BRDF from that of an ideal plane surface, may lead to an underestimation (overestimation) of the retrieved SSA ($r_{opt}$) ranging up to one order of magnitude (Kuchiki et al., 2011).



The influence of the grain shape on the SSA-albedo/BRDF relationship was explored by several authors (e.g., Jin et al., 2008; Picard et al., 2009; Libois et al., 2013). Based on ray tracing simulations, Picard et al. (2009) revealed an uncertainty of $\pm 20\,\%$ of the retrieved SSA from surface albedo measurement when the snow grain shape is unknown. These findings were in agreement with the analytical approach from the ART theory (Kokhanovsky and Zege, 2004), which accounts for parameters such as the SZA, the wavelength, $r_{\mathrm{opt}}$, SSA, and the form factor as a proxy for the shape-specific extinction property. Jin et al. (2008) studied the shape effect on satellite-based $r_{\mathrm{opt}}$-retrievals and summarized that the directional reflectance is more affected by the grain shape than the albedo, and that the best agreement to measured quantities was found when assuming aggregated snow grains. Assuming a combination of different grain shapes was proposed by Libois et al. (2015), since metamorphized snow is mostly composed of a mixture of shapes (Picard et al., 2009).

In this study, snow grain size observations derived by different methods are intercompared based on (i) in situ measurements, (ii) the application of ART theory to airborne and satellite-based remote sensing, and (iii) a minimizing approach of measured and pre-calculated top-of-atmosphere (TOA) reflectances. In contrast to previous studies on methodical comparisons (e.g., Carlsen et al., 2017), this work applies $r_{\mathrm{opt}}$ retrievals on data collected under extreme Arctic conditions (e.g., low Sun with SZA about $80°$, snow on sea ice with distinctive roughness). The results of the intercomparisons are further discussed in relation to modeled data from a numerical weather and climate model, and a parametrization of SSA evolution (Flanner and Zender, 2006). Section 2 introduces the instrumentation and analyzed data set, which was obtained in the framework of a measurement campaign in the North of Greenland during a three-week period in March/April 2018. Section 3 presents the different models used in this paper. The different retrieval methods for grain size are summarized in Section 4. The comparison of the retrieval results and simulations was done for different temporal and spatial scales. Section 5 shows this comparison, and further discusses the uncertainties caused by the surface roughness and the particle shape.

## 2 Observations

### 2.1 PAMARCMiP campaign

This study is based on measurements performed during the Polar Airborne Measurements and Arctic Regional Climate Model Simulation Project (PAMARCMiP) in 2018. PAMARCMiP 2018 belongs to a series of aircraft campaigns performed within the Arctic region (Herber et al., 2012) and was conducted together with ground-based observations from 10 March to 8 April 2018. Ground-based and airborne measurements were performed at and in the vicinity of the Villum research station (Station Nord), Greenland (81°36'N, 16°40'W) to document the short-term variability, horizontal and vertical distribution of aerosols and BC in the atmosphere, and concentrations of BC embedded in snow. The airborne activities started on 23 March 2018 and were carried out with the research aircraft Polar 5 (Wesche et al., 2016). During 14 flights cloud, aerosol (Nakoudi et al., 2020), and surface properties were quantified by in situ and remote sensing instruments. The observations mainly covered the sea ice covered Arctic ocean and the Fram Strait.

Surface properties, as the spectral surface albedo and snow grain size were derived from the spectral modular airborne radiation measurement system (SMART, Wendisch et al., 2001). The airborne laser scanner RIEGL VQ580 measured the





distance to the surface with an accuracy of about 2.5 cm (Carlsen et al., 2020). Out of these data, a 1×1 km$^2$ reference elevation model with a horizontal resolution of 1 m was generated along the flight track. The standard deviation of the relative surface elevation describes the surface roughness. A downward-looking commercial photo camera equipped with a fisheye lens was used to classify the surface conditions. To quantify atmospheric properties, dropsondes of type RD94 (Ikonen et al., 2010)

were released during the flights. Vaisala HUMICAP humidity and temperature sensors were part of the basis meteorology of the Polar 5 aircraft. An airborne sun photometer with an active tracking system (SPTA, Herber et al., 2002) was installed on the top of the aircraft and provided the aerosol optical depth (AOD) at 861 nm and 1026 nm wavelength. Atmospheric aerosol was also characterized by the Airborne Mobile Aerosol Lidar (AMALi) system (Stachlewska et al., 2010) operated in zenith-viewing direction to measure backscatter coefficient profiles at 355 and 532 nm wavelength.

## 2.2 Instrumentation to measure the snow grain size

### 2.2.1 Ground-based measurements by the IceCube system

As a ground-based reference, an IceCube instrument was used to derive the SSA during PAMARCMiP 2018. The SSA was measured on a daily basis at the ground along a fixed 100 m transect located in close vicinity of the Villum research station (distance of 2 km) between 19 March and 4 April, with about 51 samples taken each day. Additionally, broadband surface

albedo measurements were performed by a pair of stationary pyranometers installed close to this IceCube sample line. A second SSA data set was sampled between 22 March and 3 April along a 150 m transect with 5 samples each day about 600 m away from the other transect. On 28 March an extensive measurement was performed along a 15 km transect covering mainly snow-covered sea ice. With these data sets, temporal and spatial variabilities of the snow grain size and SSA within the course of the campaign were observed.

The IceCube device illuminates a snow sample with a laserdiode emitting at 1310 nm wavelength underneath an integrating sphere (Gallet et al., 2009). A photodiode detects the reflected signal, which is used to calculate the SSA based on radiative transfer simulations with an uncertainty of about 10 % for SSA values of up to 60 m$^2$kg$^{-1}$, which corresponds to a $r_{\mathrm{opt}}$ of 55 μm (Gallet et al., 2009). The limitations of the IceCube measurement principle for snow samples with smaller grain sizes is related to artefacts, which occur when the snow density is lower than 100 kg m$^{-3}$ and the radiation may reach the bottom of

the snow sample. However, the measured mean density of 230±30 kg m$^{-3}$ during PAMARCMiP 2018 did not fall below this threshold.

### 2.2.2 Airborne measurements by SMART

The SMART instrument on board of the research aircraft Polar 5 consists of optical inlets, fibre optics, spectrometers, and a data acquisition system. A set of upward and downward looking optical inlets were installed on the aircraft fuselage. The

optical inlets were actively stabilized to correct for aircraft movement (Wendisch et al., 2001). The upward and downward spectral radiation was transferred, via optical fibre, from the optical inlets to a set of four spectrometers (two for each hemisphere) covering a spectral range of 0.3 μm to 2.2 μm wavelength with a full width at half maximum of 1-–2 and 9—16 nm,





respectively (Wendisch et al., 2001; Bierwirth et al., 2009; Jäkel et al., 2013). Radiometric calibrations were performed before and after the field campaign using a NIST-certified (National Institute of Standard and Technology) radiation source (1000 W lamp). In addition, in-field calibrations were applied documenting possible temporal drifts of the SMART sensitivity during the campaign. At large solar zenith angles around 80° to 85° as present during PAMARCMiP, the uncertainty of the measured irra-

diance at flight level is increased compared to observations performed at smaller SZA. The known components of uncertainty (cosine correction, sensor tilting, absolute calibration, transfer calibration, wavelength accuracy, and dark current subtraction) of SMART were re-evaluated with respect to the large SZAs and the wavelengths applied in the snow grain size retrieval. In particular, the uncertainty of the cosine correction (4 %) and the uncertainty of the sensor tilt (2.5 %) have a major effect on the overall accuracy of the downward irradiance in the near-infrared (NIR) wavelength range, since the direct-to-global fraction is

approaching unity in this spectral range. Using Gaussian error propagation, the uncertainty of downward and upward irradiance in the NIR summed up to 5.7 % and 4.0 %, respectively.

### 2.2.3 Satellite measurements

Two different approaches for satellite SSA retrievals were considered in the study, based on data from the MODerate Resolution Imaging Spectroradiometers (MODIS) on board of the Terra and Aqua satellites and the Sea and Land Surface Temperature

Radiometer (SLSTR) instrument on board of Sentinel-3. SLSTR covers the VIS, NIR, and infrared spectral range with nine spectral channels. The used channels for the snow grain size retrievals (0.55 μm and 1.6 μm) have a spatial resolution of 500 m with a measurement accuracy between 2 and 5 % (Coppo et al., 2010). MODIS obtains data in 36 spectral channels with wavelengths ranging from 0.4 to 14.385 m. The three channels (0.47 μm, 0.85 μm, 1.25 μm) applied for the retrieval have a spatial resolution of 250 m and 500 m, respectively, and show a radiometric accuracy of 1.5 % to 3 % (Wiebe et al., 2013). For

the period and area of the PAMARCMiP observations, SLSTR data were available almost once a day, while from MODIS up to four images per day could be used for the data evaluation.

### 2.3 Measurement conditions during PAMARCMiP 2018

#### 2.3.1 Sea ice conditions

The analysis of aircraft observations focuses on the period 25–27 March 2018, when mostly cloudless conditions prevailed

along the flight paths. This restriction to a cloudless period is required in order to have collocated satellite observations of the surface available. Figure 1 shows the sea ice roughness as derived from the airborne laser scanner along the flight tracks (black lines) for these days. The lowest mean roughness was measured on 25 March with 0.18±0.11 m (max. 0.56 m), the highest on 26 March with 0.23±0.08 m (max. 0.49 m). However, the highest absolute roughness of 0.81 m was derived on 27 March. The percentage sea ice concentration derived for 26 March from satellite observations by the Advanced Microwave

Scanning Radiometer (AMSR) instrument (Spreen et al., 2008) is displayed in the background of Fig. 1 with a spatial gridding of about 3 km. While the flight track on 25 March also covered a region of sea ice concentrations down to 80 %, detected by the AMSR-2 sensor, the more northern regions overflown on the following two days were characterized by a sea ice concentration

of about 100 %. Visual camera data of the surface taken aircraft-based showed a number of open leads and young thin ice in the area overflown on 25 March. However, the snow depth maps derived from AMSR-2 measurements with a resolution of 25 km (Rostosky et al., 2018) revealed a mean snow depth averaged along the flight track changing from 22 cm to 27 cm between the flights.

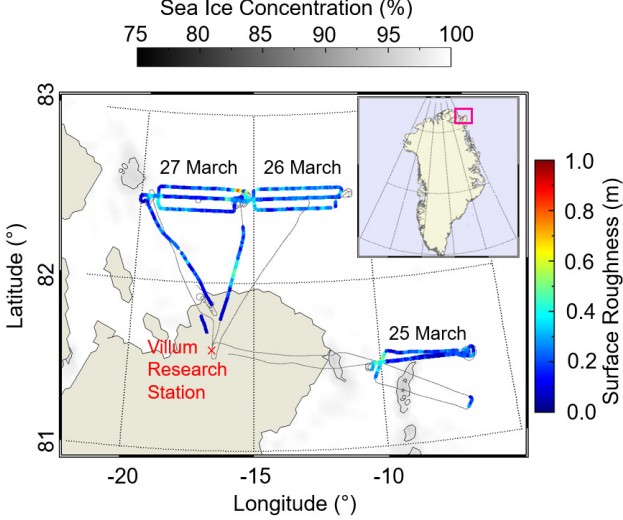

**Figure 1.** Color-coded sea ice roughness along the flight tracks for 25–27 March 2018 measured by the laser scanner on the Polar 5 aircraft. Dark gray line indicates flight track without laser scanner measurements. Sea ice concentration for this period is displayed in the background. Insert map of Greenland shows location of measurement area (red box).

### 2.3.2 Meteorological conditions

The general meteorological situation during PAMARCMiP 2018 period was characterized by a high pressure system over the North Pole and weak lows over NE Greenland, leading to a period of cloudless conditions in the measurement area between 25 and 27 March 2018. Observations of the sea ice surface properties were performed in different flight altitudes ranging from 50 m to 5 km partly passing the same location. For the radiative transfer simulations in this study the atmospheric conditions were constrained by the airborne observations. Sun photometer data were used to estimate the AOD of the entire atmospheric column. The vertical distribution of the aerosol as indicated by AOD measurements in different altitudes was similar for all three days with a continuous decrease of AOD with flight height. No indication of distinctive aerosol layers up to 5 km was given. To setup the simulations in the NIR spectral range, the AOD was extrapolated to 1100, 1280, and 1700 nm by fitting the Angstrom formula (Angström, 1964) to the measured AOD at 861 and 1026 nm. For determining the columnar AOD, only data from flight sections in the lowest altitude were taken into account. For the three flights, mean columnar AODs at 1700 nm wavelength between 0.01 and 0.03 were obtained, indicating the rather clean conditions during the three flights. In addition,





measurements with the AMALi system did not show any disturbances by clouds or aerosol layers. Only a short sequence on 25 March around 16 UTC was removed from the analysis.

The atmospheric profiles of air humidity and temperature were compiled from aircraft and dropsonde data. Dropsondes were released during the flights on 26 and 27 March, while for 25 March the atmospheric profile was derived on basis of the

aircraft meteorological sensors during a continuous ascent. The temperature profile on 26 March shows the strongest inversion of all three flights with surface temperatures of -30 °C and -10 °C around 880 hPa corresponding to an altitude of 1.1 km. The weakest inversion of about 5 K was measured on 27 March. The main flight patterns on both days were performed close to 82.5°N over sea ice. On these days, the absolute humidity below 900 hPa pressure level was significantly lower than on 25 March, when the near-surface absolute humidity was affected by areas of open water close to 81.5°N latitude. For this reason,

the largest atmospheric impact on the measured radiative quantities due to extinction is expected for the flight on 25 March.

## 3  Modeling tools

### 3.1  Snow radiative transfer model

To simulate the snow surface albedo, the open-source Two-streAm Radiative TransfEr in Snow model (TARTES) was used (Libois et al., 2013). TARTES simulates the radiative transfer in a snowpack applying the delta-Eddington approximation

(Stamnes et al., 1988). The snowpack can be constructed from a predefined number of horizontally homogeneous snow layers defined by their the snow density, specific surface area, and mass fraction of soot. The description of the single scattering properties of each layer is based on analytical equations by Kokhanovsky and Zege (2004) (see also Sec. 4).

Libois et al. (2013) and Libois et al. (2014) discussed the role of the snow grain shape on the radiative transfer in a snowpack. The grain shape is represented by the absorption enhancement parameter ($B$), and geometrical asymmetry factor ($g^{\mathrm{G}}$). $B$

accounts for the photon path length inside the snow grains due to multiple internal reflections, while $g^{\mathrm{G}}$ approximates the ratio between forward and backward scattering by the snow grains. Following Libois et al. (2013), for particles large compared to the wavelength, the asymmetry factor $g$ can be estimated by:

$$g = 0.5 \cdot (g^{\mathrm{G}} + 1) \quad . \tag{2}$$

From combining observational studies with radiative transfer modeling, Libois et al. (2013) suggested a $B$-value of 1.6 and a

value of $g$ of 0.85, representing a mixture of different grain shapes.

In the simulations of the snow surface albedo, we assumed a single snow layer without soot impurities. The snow grain size was varied between 10 μm and 300 μm. The surface albedo strongly depends on the spectral distribution and the direct-to-global fraction of the incident radiation. This input was provided by an atmospheric radiative transfer model (Sec. 3.2). Generally, the surface albedo increases with increasing SZA due to a higher probability of the photons to be scattered out of the topmost

layer of the snowpack at low Sun. Additionally, the forward scattering dominates the asymmetry of scattering, and increases the surface albedo (Warren, 1982). For low Sun, single-scattering dominates, while for higher Sun the radiation can penetrate deeper into the snowpack corresponding to a higher probability of multiple-scattering. The scattering phase function of the



snow particles depends on the snow grain shape. Therefore, the effect of the grain shape on the radiative transfer becomes more relevant for single-scattering than for multi-scattering events, when the angular scattering dependence is increasingly smeared out (Warren, 1982).

### 3.2 Atmospheric radiative transfer model

5   To calculate the direct-to-global fraction of the incident solar radiation and for the atmospheric correction of the airborne surface albedo measurements, the radiative transfer package libRadtran (Mayer and Kylling, 2005; Emde et al., 2016) was applied. As a solver for the radiative transfer equation, the Discrete Ordinate Radiative Transfer solver (DISORT) (Stamnes et al., 2000) routine was chosen. For the parametrization of the gas absorption, the SBDART model (Ricchiazzi and Gautier, 1998) was applied. The extraterrestrial spectrum was taken from Gueymard (2004). Profiles of pressure, temperature, density, 10   and gases were adapted to the airborne observations. The aerosol particle properties were specified by the spectral AOD, derived from Sun photometer measurements, the single scattering albedo ($\omega$), and the asymmetry factor of the aerosol particles. The latter two parameters were estimated from the Ny-Ålesund AERONET (AErosol RObotic NETwork) data set with $\omega = 0.95$ and $g = 0.65$ in the NIR. The impact of the uncertainty of $\omega$ and $g$ on the simulated NIR spectral irradiance is low, since the AOD derived for the selected data set did not exceed 0.03 (Sec. 2). Simulations using an $\omega$ of 0.99 and a $g$ of 0.58 resulted in 15   a difference to the default settings of less than 1 %.

### 3.3 Weather and climate model

The ICOsahedral Nonhydrostatic model (ICON, Zängl et al., 2015) is a weather and climate model developed by the German Weather Service (DWD) and Max Planck Institute for Meteorology (MPI-M). The model system solves the compressible Navier-Stokes equations on an icosahedral grid, which can be seamlessly adjusted in resolution for global and regional simu- 20   lations. A detailed description of the model can be found in Zängl et al. (2015) and Giorgetta et al. (2018). With the extension Aerosols and Reactive Trace gases (ICON-ART) developed at the Karlsruhe Institute of Technology (KIT), the model is able to simulate aerosols, trace gases, and related feedbacks (Rieger et al., 2015; Schröter et al., 2018). The limited area mode, applied here, enables the model to simulate a confined region at high resolution with prescribed lateral boundary conditions. The simulation was run with a horizontal resolution of approximately 3.3 km. The initial state and the boundaries were driven with data 25   from the Integrated Forecasting System (IFS) of the European Centre for Medium-Range Weather Forecasts (ECMWF) and fed in at six hours intervals. ICON currently has two different snow models. The first is a single-layer snow model used for the operational weather forecast. The second is an experimental multi-layer snow model (Machulskaya and Lykosov, 2008), which was applied in this study in a three layer set up. To investigate the impact of aerosols on the optical properties of snow, the model was extended by the optical-equivalent snow grain radius as a new prognostic variable, whereby the aging is based on 30   Essery et al. (2001). The acceleration of growth due to rain was additionally added and the reduction by snowfall was adjusted.





### 3.4 Parametrization of SSA evolution

Flanner and Zender (2006) parameterized the SSA evolution of dry snow with respect to the effect of the local temperature gradient and the curvature growth following the approach by Legagneux et al. (2004). Based on observational data they proposed an empirical relation of temporal SSA evolution and temperature controlled by the fit parameters $\kappa$ and $\tau$:

$$\text{SSA}(t) = \text{SSA}_0 \cdot \left( \frac{\tau}{t+\tau} \right)^{1/\kappa} , \tag{3}$$

with $t$ for time and $\text{SSA}_0$ representing the initial SSA at $t = 0$. Simulations in this study were performed for a set of parameters $\tau$ and $\kappa$ representative for a range of snow temperatures (-37°C to -28°C) and vertical temperature gradients (0 K cm$^{-1}$ to 0.5 K cm$^{-1}$). The temperature dependent best-fit-parameters for $\tau$ and $\kappa$ were fitted to adapt them to the temperature range during the considered period.

### 4 Snow grain size retrieval methods

#### 4.1 XBAER retrieval of snow grain size using satellite-based Sentinel-3 data

The eXtensible Bremen Aerosol/cloud and surfacE parameters Retrieval (XBAER) algorithm is a generic algorithm, which can derive aerosol (Mei et al., 2017), cloud (Mei et al., 2018), and surface (Mei et al., 2020a) properties from satellite observations. It has recently been extended to derive snow grain size, snow particle shape, and SSA using the Sea and Land Surface Temperature Radiometer (SLSTR) instrument on board Sentinel-3. The retrieval process is performed utilizing a Look-Up-Table (LUT). In the LUT, snow optical properties are pre-calculated for nine pre-defined ice crystal particle shapes (aggregate of 8 columns, droxtal, hollow bullet rosette, hollow column, plate, aggregate of 5 plates, aggregate of 10 plates, solid bullet rosette, column; Yang et al., 2013). An atmospheric correction step is applied with a weakly absorbing aerosol type (Mei et al., 2020a) and AOD from Modern-Era Retrospective Analysis for Research and Applications (MERRA) simulation. The aerosol profile is approximated by a exponential function between surface and 3 km altitude. Other trace gas profiles, are taken from a monthly latitude-dependent mean climatology. Snow grain size and snow particle shape are then obtained by minimizing the surface directional reflectances at two wavelengths (0.55 μm and 1.6 μm), between theoretical simulations and SLSTR observations. The sensitive study, as presented in Mei et al. (2020b), shows that the impact of snow particle shape selection on the the $r_{\text{opt}}$-retrieval is significant, and potential cloud/aerosol contamination introduce an underestimation of $r_{\text{opt}}$. The comparison between XBAER derived snow grain size and ground-based measurements shows a relative difference of less than 5 % (Mei et al., 2020c).

#### 4.2 SGSP retrieval of snow grain size using satellite-based MODIS data

In this study the snow grain size and pollution amount (SGSP) retrieval algorithm by Zege et al. (2011) was applied to MODIS data. Following Wiebe et al. (2013), the SGSP retrieval does not require a-priori information on the snow grain shape. Radiances of MODIS measured in three channels (469 nm, 858 nm, and 1250 nm) are used in this method, which reveals a snow grain size





retrieval uncertainty of 10 % for SZA lower than 75° (Zege et al., 2011). This uncertainty increases up to 20 % for SZA = 85°
(Carlsen et al., 2017).

The SGSP retrieval method uses an analytical asymptotic solution of the radiative transfer equation (Kokhanovsky and
Zege, 2004). Following Zege et al. (2011), the plane surface albedo $\alpha_\mathrm{p}(\theta_0)$, corresponding to the hemispherical reflectance and
assuming only direct illumination, can be calculated as a function of the solar zenith angle $\theta_0$ by:

$$\alpha_\mathrm{p} = \exp[-y \cdot K_0(\theta_0)] \quad , \tag{4}$$

where $K_0$ represents the escape function determining the angular distribution of radiation, which escapes from a semi-infinite,
non-absorbing medium as approximated by (Kokhanovsky, 2003) with:

$$K_0 = \frac{3}{7} \cdot [1 + 2\cos(\theta_0)] \quad . \tag{5}$$

For completely diffuse illumination Eq. (4) reduces to:

$$\alpha_\mathrm{s} = exp[-y] \quad , \tag{6}$$

defining the spherical albedo $\alpha_\mathrm{s}$ (Zege et al., 2011). According to Kokhanovsky and Zege (2004) and Zege et al. (2011), $y$ in
Eqs. (4) and (6) can be written as:

$$y = A \cdot \sqrt{\frac{4\pi \cdot \chi(\lambda)}{\lambda} \cdot r_\mathrm{opt}} \quad , \tag{7}$$

when considering radiative transfer in a dense snowpack, with $\chi$ being the imaginary part of the complex refractive index of
ice, wavelength $\lambda$, which is taken from Warren and Brandt (2008). $A$ represents the form factor, which depends on the particle
shape, and combines the absorption enhancement parameter $B$ and the asymmetry parameter $g$:

$$A = \frac{4}{3}\sqrt{\frac{2B}{1-g}} \quad . \tag{8}$$

Zege et al. (2011) gave a range of $A$ between 5.1 for fractals (Kokhanovsky and Macke, 1997) and 6.5 for spheres. This range
of possible values of $A$ contributes to the uncertainty of the retrieved $r_\mathrm{opt}$ (25 %) due to the unknown particle shape. The SGSP
retrieval uses an averaged value for $A$ of 5.8 with $B = 1.5$ and $g = 0.84$, derived for a mixture of randomly oriented hexagonal
plates and columns. To reduce uncertainties by using different settings for the satellite retrieval and the TARTES simulations,
we set $A = 5.8$ in both applications.

Since satellites cannot measure the albedo directly to relate the snow albedo and the snow grain size using Eq. (4), further
assumptions are applied in the SGSP retrieval. In general, the BRDF is used for albedo calculations. The snow grain size is
much larger than the wavelength, therefore the geometrical optics holds and the scattering phase function of the snow grains is
defined by the particle shape and the real part of the refractive index, which shows no significant wavelength-dependence for
the three MODIS channels used in the SGSP retrieval. Consequently, also the phase function and the BRDF can be assumed
to be spectrally independent (Zege et al., 2011). Satellite-based measurements of the snow surface reflectance are determined





by both atmospheric and surface contributions. The SGSP retrieval accounts for the surface contribution by employing the radiative transfer model RAY (Tynes et al., 2001) using the subarctic winter atmospheric model (Kneizys et al., 1996) and the Arctic background aerosol model (Tomasi et al., 2007). By considering the atmospheric contribution and assuming the spectral independence of the BRDF, the optical grain size is determined iteratively (Zege et al., 2011; Wiebe et al., 2013). Further

details regarding to the theoretical background of the SGSP retrieval and the applied equations were given in Zege et al. (2011) and Wiebe et al. (2013).

### 4.3    Snow grain size retrieval using airborne SMART data

#### 4.3.1    Retrieval method

In this study we applied the TARTES model to derive the snow grain size from aircraft observations, which gives flexibility to

adjust the snow settings. The retrieval of snow grain sizes from the airborne spectral albedo measurements is based on Carlsen et al. (2017). They applied a modified approach from the SGSP retrieval by Zege et al. (2011) using the albedo ratio ($\mathcal{R}$) of the SMART measurements at $\lambda_1 = 1280\,\text{nm}$ and $\lambda_2 = 1100\,\text{nm}$ wavelength. Based on Eqs. (4) and (7) they related the snow grain size to $\mathcal{R}$ by:

$$r_{\text{opt}} = \left\{ \frac{\ln \mathcal{R}}{A \cdot K_0(\theta_0) \cdot \left[ \sqrt{\frac{4\pi \cdot \chi(\lambda_2)}{\lambda_2}} - \sqrt{\frac{4\pi \cdot \chi(\lambda_1)}{\lambda_1}} \right]} \right\}^2 \tag{9}$$

to minimize the retrieval uncertainty, which is affected by the measurement uncertainty of the spectral albedo. These aircraft measurements were performed over the Antarctic Plateau under clean atmospheric conditions, such that the direct-to-global fraction ($f_{\text{dir/glo}}$) in the NIR spectral range is close to unity. This allows the usage of the plane albedo for the $r_{\text{opt}}$-retrieval as done by Carlsen et al. (2017). However, for the conditions during PAMARCMiP, also the diffuse incident radiation needs to be considered, as the airborne measurements correspond to the blue sky albedo ($\alpha_{\text{b}}$). The blue sky albedo can be understood as a

linear combination of the plane and spherical albedo:

$$\alpha_{\text{b}} = \alpha_{\text{p}} \cdot f_{\text{dir/glo}} + \alpha_{\text{s}} \cdot (1 - f_{\text{dir/glo}}) \quad . \tag{10}$$

To estimate $r_{\text{opt}}$ from Eq. (10), a nonlinear least square method is applied, which minimizes the root mean square deviation between the measured and modeled albedo, accounting for $f_{\text{dir/glo}}$. Different to Carlsen et al. (2017), in this study TARTES simulations were performed together with libRadtran calculations to generate LUTs accounting for the specific atmospheric

conditions during the PAMARCMiP aircraft observations. Both, TARTES and the SGSP retrieval method, rely on the same theoretical background based on formalism by Kokhanovsky and Zege (2004). Figure 2 compares the dependence of snow surface albedo with snow grain size for the different approaches. Neglecting the diffuse incident contribution for the PAMARCMiP conditions, would result in a significant difference of the calculated surface albedo for SZA $= 80°$ and A $= 5.8$ (Fig. 2). For all wavelengths, the parameterized plane albedo (dashed lines) using Eq. (4) is larger than the results from the TARTES simula-

tions (solid lines) and blue-sky-albedo calculations applying Eq. (10) (filled squares), which account for the proper $f_{\text{dir/glo}}$.



The direct-to-global fraction and consequently the offset between the plane and blue-sky-albedo are wavelength-dependent, such that $\mathcal{R}$ shows also a bias between both methods. This indicates the need for considering the direct to global fraction in the retrieval.

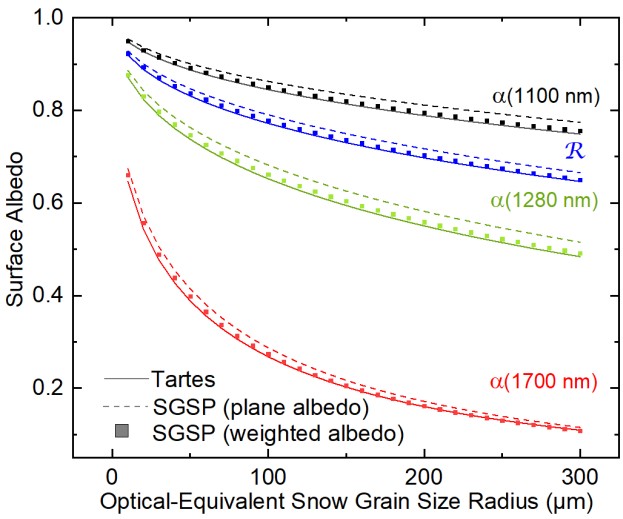

**Figure 2.** Simulated surface albedo of different snow grain sizes $r_{\mathrm{opt}}$ for 1100, 1280, and 1700 nm wavelength using TARTES (solid lines) and the SGSP equations for plane and blue-sky-albedo (dashed lines, filled squares) for SZA = 80°. Additionally, the albedo ratio $\mathcal{R}$ is plotted for all methods.

### 4.3.2 Atmospheric correction

The albedo parametrization used in Sec. 4.3.1 is valid for observations at the surface but might be biased when the spectral albedo is measured in flight altitude. Scattering by atmospheric constituents increases the albedo in flight altitude compared to the surface albedo. Therefore an atmospheric correction was applied following the method by Wendisch et al. (2004). It is based on an iterative algorithm, which uses radiative transfer simulations applying the radiative transfer package libRadtran (Mayer and Kylling, 2005; Emde et al., 2016).

The atmospheric masking over Arctic snow can contribute to significant uncertainties in the albedo-based $r_{\mathrm{opt}}$-retrieval. The atmospheric effects representative for the PAMARCMiP conditions are illustrated in Fig. 3. The spectral surface albedo was simulated for snow grain sizes between 60 μm and 350 μm (SSA: 9.3 to 55 m² kg⁻¹) using the TARTES model (gray scaled solid lines in Fig. 3). The spectral surface albedo for $r_{\mathrm{opt}} = 60$ μm was set as input for atmospheric radiative transfer simulations with libRadtran to calculate the upward and downward spectral irradiances at 200 m and 3 km altitude, corresponding to
common flight altitudes during PAMARCMiP. The height-dependent albedo calculated from the simulated irradiance spectra are shown as dotted and dashed red lines in Fig. 3. The two major absorption bands of water vapour are marked by gray bars. However, the effect of extinction by atmospheric constituents can be observed also outside the marked gray areas. Over bright

surfaces, such as snow, the atmospheric masking results in a reduction of the albedo in higher altitudes compared to the surface albedo as shown in this example. For the $r_{\mathrm{opt}}$-retrieval wavelengths 1100 nm and 1280 nm (both indicated by a vertical line in Fig. 3) as used by Carlsen et al. (2017), the albedo at 3 km altitude shows a reduction of 0.14 and 0.12, respectively, at both wavelengths as compared to the default surface albedo. The atmospheric impact on the albedo for 200 m flight altitude is rather

small with a bias of -0.01. However, the bias would directly contribute to a $r_{\mathrm{opt}}$-retrieval error, if no atmospheric correction was applied. The snow grain size matching with the uncorrected albedo at 1280 nm wavelength at 3 km altitude, for example, would result in an overestimated $r_{\mathrm{opt}}$ of about 150 μm (SSA = 22 $\mathrm{m^2\,kg^{-1}}$) compared to the default 60 μm snow grain size. This clearly demonstrates the relevance of the atmospheric correction when using wavelengths, which are highly affected by water vapour absorption.

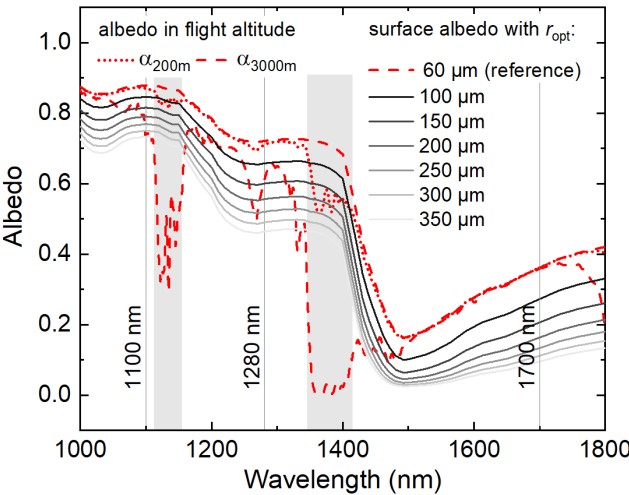

**Figure 3.** Spectral albedo derived from simulated upward and downward irradiance spectra at different altitudes (2000 m, 3000 m) over the surface (red dotted and red dashed line). The default surface albedo (thick solid red line) represents the input surface albedo with a $r_{\mathrm{opt}} = 60$ μm (derived from TARTES simulations). The other solid lines are simulated albedo spectra for $r_{\mathrm{opt}}$ ranging between 100 μm and 350 μm. The shaded areas mark the absorption bands of water vapour.

### 4.3.3    Wavelength choice and penetration depth

The atmospheric corrections reduces with increasing wavelength due to the reduced scattering by atmospheric aerosol particles and molecules. Therefore, the retrieval of the surface albedo from the airborne measurements is less affected by the quality of the atmospheric correction for larger wavelength. To make use of this reduced uncertainty, we shifted the retrieval wavelength of the snow grain size to the wavelength at 1700 nm. At this wavelength, the atmospheric effect on the albedo for PAMARCMiP

conditions can be neglected, as the surface albedo and the albedo derived in 3 km are almost identical (Fig. 3). However, using different retrieval wavelengths might result in different $r_{\mathrm{opt}}$-estimates because the penetration depth of the radiation in the snow depends on the wavelength and therefore weights the vertical structure of the snowpack differently. This becomes crucial





if the snow layers are stratified, such that a vertical difference in the snow grain size can impose systematic differences in the retrieval. According to the Beer-Lambert law, the radiation decreases exponentially with penetration. The distance in the snowpack where the incident irradiance has decayed to $1/e \sim 37\,\%$ of its value is the $e$-folding depth ($z_e$). It is used as a measure to quantify, for which layer the retrieved snow grain sizes are representative. For snow, the $e$-folding depth is calculated by:

$$
5 \quad z_e = \left\{ 3 \cdot \frac{\rho_{\mathrm{snow}}}{\rho_{\mathrm{ice}}} \cdot \sqrt{2\pi \cdot \frac{\chi(\lambda)}{\lambda \cdot r_{\mathrm{opt}}} \cdot B \cdot (1-g)} \right\}^{-1} \quad , \tag{11}
$$

following Zege et al. (1991), with $\rho_{\mathrm{snow}}$ and $\rho_{\mathrm{ice}}$ representing the densities of ice and snow. The penetration depth increases with decreasing wavelength and snow density, as well as with increasing snow grain size, i.e. decreasing optical depth of the snow layer. From the ground-based snow measurements during PAMARCMiP, snow densities between 200 kg m$^{-3}$ and 300 kg m$^{-3}$ and snow grain sizes around 60 µm were derived. For these conditions and the wavelength range between 1100 nm and 1700 nm, the snow grain size retrieval refers to snow layers between 0.07 cm and 0.62 cm. Assuming that $r_{\mathrm{opt}}$ does not change significantly within the first cm of the snowpack, the choice of wavelength to derive $r_{\mathrm{opt}}$ from surface albedo data, is of minor importance. Snow pit measurements of the snow grain size and the snow density in the vicinity of the Villum research station have shown only a low variability (less than 5 µm difference) within the first 10 cm of the snowpack, which supports the approach using wavelengths between 1100 nm and 1700 nm for the $r_{\mathrm{opt}}$-retrieval.

15 **4.3.4 SMART measurement uncertainty and retrieval sensitivity**

The accuracy of the snow grain size retrieval is affected by uncertainty of the surface albedo or the albedo ratio, respectively. The total uncertainty of the surface albedo retrieved form the airborne observations is estimated to be about 7.1 %. By using the albedo ratio $\mathcal{R}$, the uncertainty reduces to 5.8 % as the transition to relative measurements provides independence from the absolute calibration. The uncertainty of the retrieved snow grain size is affected by the measurement uncertainty of the surface albedo.

To estimate the contribution of the SMART measurement uncertainty on the accuracy of the snow grain size retrieval, the measurement uncertainties were propagated through the retrieval algorithm, exemplary for a SZA of 80°. Figure 4a compares the true snow grain size (without variation) and the uncertainty range of the retrieved snow grain size including variation in both directions ($\pm \Delta \alpha$). For larger grain sizes, the relative uncertainty of the retrieval increases as the sensitivity of the surface albedo to changes of the grain size becomes smaller. A higher albedo causes a lower retrieved snow grain size, which corresponds to the region with a larger slope of the LUT (Fig. 2). The relation between snow grain size and surface albedo for 1700 nm wavelength shows a stronger decrease than using $\mathcal{R}$. This leads to a higher sensitivity on the SMART measurement uncertainty applying $\mathcal{R}$, and an absolute error, which is about three to five times higher than using $\alpha(1700\,\mathrm{nm})$ for the snow grain size retrieval. The relative bias (Fig. 4b) clearly demonstrates the snow grain size dependence of the uncertainty. For small grain sizes, as those of fresh fallen snow, the retrieved snow grain size could be overestimated by about 100 % when applying the ratio method, while the other approach would lead to uncertainties less than 25 % for all grain sizes.



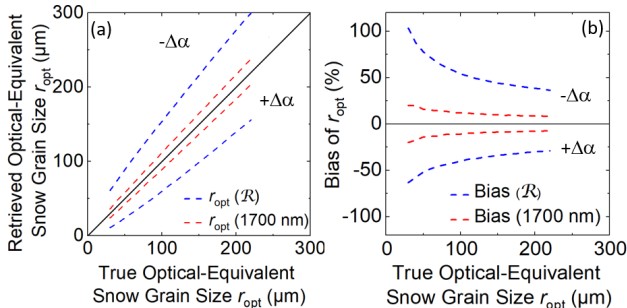

**Figure 4.** Estimated snow grain size retrieval uncertainty due to SMART measurement uncertainty for SZA = 80°. (a) Correlation of retrieved and true $r_{\mathrm{opt}}$, (b) bias in % depending on true $r_{\mathrm{opt}}$.

## 5   Comparison of retrieval results

### 5.1   Temporal variability: local observations and modeling

Daily ground-based snow grain size measurements near the Villum research station were performed during PAMARCMiP over almost three weeks starting on 19 March 2018. At the beginning of the measurement period a hard crust covered with

only some millimeter of snow was observed, which resulted from a refreezing period after a massive snow melting event in the end of February 2018. After days of snowfall, a period of dry and mostly cloudless conditions followed, whereby the air temperature did not exceed -25 °C. The spatially averaged snow grain size data along line A (100 m, 51 samples) and along line B (150 m, 5 samples) are shown in Fig. 5. The error bars indicate the 1-sigma standard deviation calculated from the total set of samples. In general, $r_{\mathrm{opt}}$ increased slightly at both sample locations between 44 μm and 72 μm within the three weeks

of measurements. The highest variability was observed in the first period of snowfall up to the onset of the cloudless period on 25 March 2018. The day-to-day variation in this first week of observations was stronger than for the following periods, and the spatial variation between the 51 samples along the 100 m transect (line A) covered almost the entire range of $r_{\mathrm{opt}}$-values of the three weeks of measurements. Weak snowfall and blowing snow were reported on 20 March, drifting snow and weak snowfall for 22 March, which might explain the striking variability on these two days. The spatially averaged snow grain size

along line B showed mostly higher $r_{\mathrm{opt}}$-values than measured along line A (600 m away), in particular in the first week with snowfall and drifting and blowing snow conditions. For the remaining period, both data sets agreed within the range of the individual standard deviations. The range and the temporal evolution of the measured snow grain size is less strong, with an increase of 15 μm within five days after snowfall, than observed by Carlsen et al. (2017) for example. Their measurements on the Antarctic Plateau have shown a more pronounced daily increase in snow grain size after snowfall of about 5.8 μm day$^{-1}$

(daily SSA decrease: 3.2 m$^2$ kg$^{-1}$ day$^{-1}$) under a similar temperature regime (-20 °C to -35 °C).

The snow grain size evolution simulated with ICON-ART is shown in Fig. 5 as solid line. The simulation assumes a growth rate factor of 0.06 μm$^2$ s$^{-1}$ for this temperature and snow grain size range as suggested in Essery et al. (2001). The snowfall period before 25 March is well covered by the ICON-ART simulations. However, the growth of the snow particles evolves

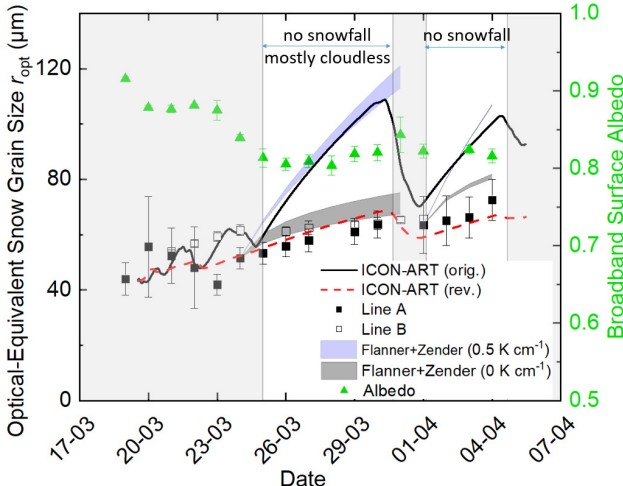

**Figure 5.** IceCube measurements of the snow grain size during PAMARCMiP at two locations (along line A and line B) in the vicinity of the Villum research station. Bars give the standard deviation of the spatial averaging. Mean broadband surface albedo derived from ground-based pyranometer data near line A is additionally shown (triangles). ICON-ART modeled snow grain sizes are indicated as black solid and red dashed lines. Spread of parametrization result of the snow grain size evolution after Flanner and Zender (2006) is shown as dark gray and light blue areas.

rapidly in the cloudless period reaching $r_{opt}$-values up to 110 µm (SSA = 30 m$^2$ kg$^{-1}$) which is about twice the numbers derived from the IceCube measurements. With the onset of the second short period of snowfall on 30 March, the snow grain size decreased to a value similar to the in situ observations. The comparison shows, that the snow grain size of new snow can be well reproduced by ICON-ART. However, the aging process is not well represented by the growth rate factor from Essery et al.

(2001) for the specific conditions during PAMARCMiP. Therefore, the parametrization of the growth rate factor in ICON-ART was adjusted, such the simulated snow grain size covers the in situ measurements (red dashed line in Fig. 5). For the specific temperature and snow grain size during PAMARCMiP, the growth rate factor was reduced to about 0.012 µm$^2$ s$^{-1}$, one fifth of the original value.

In addition to ICON-ART, the parametrization by Flanner and Zender (2006) was compared to the observations. For the

10 precipitation-free period starting at the end of 24 March we calculated the snow grain size evolution based on Eq. (3) for two scenarios with d$T$/d$z$ = 0 K cm$^{-1}$ and d$T$/d$z$ = 0.5 K cm$^{-1}$. From snow pit measurements performed on 24 March, a vertical temperature gradient of about 0.4 K cm$^{-1}$ was derived. Further snow temperature measurements were only conducted at the top of the snowpack covering a range between -28 °C and -37 °C. The dark gray areas shown in Fig. 5 display the range of the parameterized snow grain size increase assuming no vertical temperature gradient when particle evolution is dominated by the

15 curvature growth. For this scenario, the snow grain size was only slightly overestimated by the parametrization, compared to the in situ measurements. In contrast, a significant overestimation was observed assuming a vertical temperature gradient of 0.5 K cm$^{-1}$ (blue shaded areas in Fig. 5). The snow grain size for this scenario matches well with the ICON-ART simulations.





However, the vertical temperature gradient effect, which leads to a faster snow metamorphism in terms of a faster snow grain size increase, seems to be less relevant than calculated. One of the reasons for this poor representation of snow grain size evolution might be caused by the fitting of the temperature-dependent parameters, $\tau$ and $\kappa$, reported in Flanner and Zender (2006) to the temperature range in this study. For $dT/dz = 0\,\mathrm{K\,cm^{-1}}$, the original parameters were described by an exponential

decay fitting with a coefficient of determination ($R^2$) larger than 0.99, while for $dT/dz = 0.5\,\mathrm{K\,cm^{-1}}$ the fitting quality was significantly lower [$R^2(\tau) = 0.5$, $R^2(\kappa) = 0.99$]. The sensitivity of $\tau(T)$ and $\kappa(T)$ to temperatures in the range between -20 °C and -50 °C (see Tab. 4 in Flanner and Zender, 2006), differs from the parametrization for the range between -20 °C and 0 °C, resulting in this lower fitting quality. This likely is caused by the suppression of the temperature gradient effect at -50°C, which is strongly affecting the parametrization between -20 and -50°C.

The snow metamorphism affects the measured broadband surface albedo, which is shown in Fig. 5. The broadband surface albedo derived from the ground-based pyranometer measurements were calculated from daily averages of measurements between 15:30 and 16:30 UTC, when daily IceCube measurements were carried out. The highest surface albedo values were observed before 25 March. With the onset of the cloudless period the broadband surface albedo decreased from about 0.87 to about 0.80. Radiative transfer simulations were applied to estimate the surface albedo for the conditions on 22 March and 26

March, representing the overcast (period I) and cloudless period (period II). While period I (22 March) was characterized by a SZA of 83 °, and a $r_{\mathrm{opt}}$ of 50 µm, the parameters of period II (26 March) were estimated with SZA = 81.4 ° and $r_{\mathrm{opt}}$ = 60 µm. The simulations revealed differences of 0.09 between period I ($\alpha$ = 0.96) and period II ($\alpha$ = 0.87), which are higher than observed from the pyranometer measurements (0.07).

The observed decrease of broadband surface albedo during PAMARCMiP might be caused by the increase of snow grain

size and/or the change of the atmospheric conditions. To separate these two effects, period II was recalculated assuming a $r_{\mathrm{opt}}$ of 50 µm (as in period I), which resulted in an increase by 0.01 and consequently indicated only a minor effect by the snow grain size variation. For a more detailed investigation of the atmospheric impact on the broadband surface albedo, period II was re-simulated using the spectral surface albedo from TARTES, which assumed only a diffuse component (spherical albedo) due to the impact of the cloud layer from period I. The new setup forced the surface broadband albedo to increase by 0.08

emphasizing the impact of clouds. Consequently, for the discussion on the snow grain size effect on the surface albedo, the atmospheric impact has to be separated, such that the temporal decrease of the surface albedo in Fig. 5 was attributed to the cloud impact rather than to the increase of the snow grain size for the PAMARCMiP period and conditions.

## 5.2 Spatial variability: Airborne and satellite observations

### 5.2.1 Retrieved maps of snow grain size

Maps of the retrieved snow grain size from the SGSP and XBAER retrieval techniques using satellite data, as well as the reflectance at 1.24 µm wavelength from MODIS measurements at 11:50 UTC for 25 March are shown in Fig. 6. The snow grain size estimated from the SMART measurements along the flight track (14 - 17 UTC) are displayed on each of the panels and were calculated using TARTES generated LUTs based on the 1700 nm surface albedo. Four MODIS overpasses were





evaluated for the period and region of aircraft observations on this day. The different number of valid data points led to an irregular spatial distribution of the snow grain size in each of the four MODIS maps (Fig. 6a-d). Note, that the ICON-ART simulations were restricted to areas where the model-based land mask indicated land surfaces only, and are not included in the analysis here.

**Figure 6.** Snow grain size estimated from MODIS satellite observations and applying the SGSP retrieval (a) - (d), and XBAER retrieval results on SLSTR data (e) for the area overflown with the Polar 5 aircraft on 25 March 2018. (f) MODIS reflectance measured at 1.24 μm wavelength. The color-coded $r_{opt}$ derived from SMART measurements (1700 nm - method) is overlaid. The 4 numbers and reddish points indicated in (f) refer to specific types of surfaces (see Fig. 9).





As illustrated in Fig. 6, the main spatial features of the retrieved snow grain size show similar patterns from west to east with lowest $r_{opt}$-values over land, increasing $r_{opt}$-values near the eastern coast of Greenland, an area of slightly decreasing $r_{opt}$ (near 9°W), and highest values in the most eastern part of the overflown area. Both, satellite and airborne observations, revealed less variation of the snow grain size over Greenland than over the sea ice. Over Greenland, the retrieved $r_{opt}$ was mostly less

than 100 µm (SSA = 33 m² kg⁻¹) , while $r_{opt}$ over sea ice reached values of up to 300 µm (SSA = 11 m² kg⁻¹). An exception was found for the map from the MODIS 16:45 UTC overpass, where significant lower $r_{opt}$-values were retrieved over the sea ice (Fig. 6b). At this time the SZA ranged between 82.4° and 84° for the entire scene. The SZA of the other satellite overpasses were smaller between 79.1° and 81.9°. As discussed earlier, the retrieval uncertainty increases with increasing SZA, which might be one of the reasons for the apparent different spatial snow grain size pattern observed in the late afternoon overpass

(Fig. 6b). The spatial distribution of the reflectance at 1.24 µm wavelength (Fig. 6f), which is sensitive to the snow grain size, shows an increasing surface inhomogeneity in the eastern region with the highest $r_{opt}$-values. A low reflectance at this wavelength does not necessarily correspond to open water. It might also indicate young ice areas with a possible thin snow layer on top, which causes an overestimation of the derived snow grain size. For example, in the area centered at 81°N/11°W, such low reflectances together with high $r_{opt}$-values were measured, while the AMSR instrument indicates a closed sea ice

cover. Note, that the measurements might be affected by thin low level clouds generated above open leads, which were not completely excluded from the data analysis.

Limited to the area of the Sentinel-3 overpass, the frequency distributions of $r_{opt}$ are shown for each overpass of MODIS and SLSTR in Fig. 7. The 13:50 UTC overpass was excluded in this analysis due to the high amount of unclassified pixel, which would bias the statistics of this case. The plot of the relative frequencies (in $r_{opt}$-bins of 10 µm) shows two main modes

for the three MODIS-based distributions. These two modes represent the lower snow grain sizes over land and the higher numbers retrieved over sea ice. The two morning overpasses revealed similar distributions over sea ice, but some shift of the "land"-mode by 20 µm snow grain size. Corresponding to Fig. 6b the relative frequency of the MODIS data from 16:45 UTC revealed the smallest distribution and the smallest $r_{opt}$-values compared to the other MODIS overpasses. However, the XBAER retrieval shows a significant smaller variability with a more narrow frequency distribution. A narrow mode with a maximum

at 120 µm marks the snow grain size derived over land. Over sea ice, there are two further modes (maxima at 140 µm and 180 µm, respectively), with the third mode resulting from the highest $r_{opt}$-values measured over the most Eastern region, where the surface is more heterogeneous (Fig. 6f).

### 5.2.2 Statistical comparison

The statistical measures of the retrieved snow grain size are summarized in Fig. 8. For the 25 March, the data was split into

observation above sea ice and land, where also ICON-Art simulations were available. The satellite data were matched to the flight track of the Polar 5 aircraft before the statistical mean, the median, the first and third quartile, and the minimum and maximum values without outliers were calculated. A running average of the SMART measurements was used to account for the spatial resolution of the satellite data. For the 25 March (sea ice), the analysis reveals that the interquartile ranges (IQR), indicated by the gray boxes, cover different $r_{opt}$-ranges, especially for the SGSP retrievals of the MODIS data from 13:50 UTC



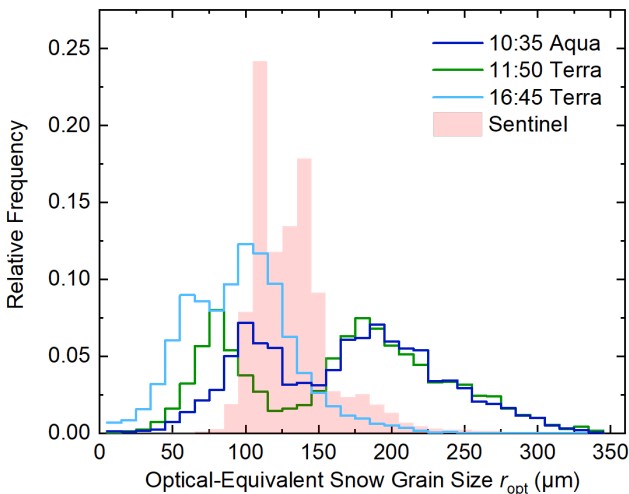

**Figure 7.** Relative frequency of the retrieved snow grain size from Sentinel-3, Aqua, and Terra overpasses on 25 March 2018.

and 16:45 UTC. The applied SGSP retrieval exhibit no clear bias compared to the other methods as can be concluded from the broad range of retrieved snow grain sizes. The XBAER retrieval shows the smallest IQR, and apart from the 16:45 UTC MODIS overpass, also the lowest mean $r_{opt}$. For the SMART albedo measurements both, the ratio method, and the TARTES-generated LUTs for 1700 nm were applied, revealing differences of the mean $r_{opt}$-value of 45 μm for the flight over sea ice, and 12 μm for snow measurements over land. This corresponds to differences in the SSA of about $5\,\text{m}^2\,\text{kg}^{-1}$ over land and sea ice.

Overall, the spread between the mean $r_{opt}$-values of the different methods is significantly lower over the land surface than over the sea ice (Fig. 8). Apart from the XBAER retrieval, the qualitative differences between the methods is similar for observations over land and sea ice, with lowest (highest) $r_{opt}$-values for the 16:45 UTC (13:50 UTC) observations by MODIS. The best agreement to IceCube measurements ($r_{opt}$ = 53 μm) on this day were derived from the 16:45 UTC MODIS overpass ($r_{opt}$ = 54 μm). All other retrievals and simulations showed a positive bias compared to the in situ measurements. XBAER and the SGSP retrieval of the 13:50 UTC MODIS overpass deviate almost by a factor of three from the IceCube results. Possible reasons for the deviation of the XBAER results, but also of the SMART retrieval, could originate from the assumption of the snow particle shape when calculating the LUTs. While for the SMART retrieval, a mixture of grain shapes is assumed, XBAER estimates the snow grain shape in 9 classes (Mei et al., 2020b) in addition to the snow grain size. For the considered area, mostly droxtals were retrieved over the land and the coastal region, and aggregates of 8 columns over sea ice. The use of an inappropriate ice crystal shape in XBAER leads to an error of less than 10% to more than 50% in the retrieval of grain size, depending on the grain size value itself (Mei et al., 2020b). The similar treatment of ice crystal shape in the SGSP and SMART retrieval provides a similar retrieved grain size while a totally different ice crystal shape assumption in the XBAER algorithm leads to a larger diversity, compared to the SGSP and SMART results. An independent ground-based measurement data set will provide, in the future, a better understanding of the similarity and diversity between different satellite retrievals.



**Figure 8.** Box-and-whisker plot of snow grain size statistics along the entire flight track over sea ice (a) and over Greenland (b) on 25 March 2018, as well as over sea ice on 26 March 2018 (c), and on 27 March 2018 (d) . Minimum and maximum values (without statistical outliers) are displayed as vertical bars. The boxes indicate the 25th, 50th (median), and 75th percentiles of the distribution. IQR stands for interquartile range. The mean $r_{opt}$ from the IceCube is plotted as dashed horizontal line in (b).

The snow grain size statistics for 26 and 27 March are displayed in Fig. 8c, d. The region observed during both flights was further north than on 25 March (about 82.5°N), and AMSR-2 data covered only sea ice with concentrations of about 100 % (Fig. 1). Due to the earlier flight time (12 UTC on 26.3. and 13 UTC on 27.3.), the SZA was in a similar range compared to 25 March. The satellite overpasses on these two days were also within 79° and 82° SZA. Comparing the statistics of the three flights, the lowest variability between the methods was found for 26 and 27 March flights, which were characterized by a closed sea ice cover. However, the mean snow grain size along the flight tracks may still vary by 100 μm (SSA variation:





$25 \, \mathrm{m^2 \, kg^{-1}}$), which corresponds to a bias of about $100\,\%$. The retrieval results of the MODIS instruments suggest that the snow grain size from the Terra satellite tends to be lower and shows less variation than the $r_{\mathrm{opt}}$-distribution derived from the Aqua satellite. Comparing the MODIS snow grain size within a 10 km radius around the Villum research station with IceCube measurements (53 - 58 µm), revealed best agreement with $r_{\mathrm{opt}}$-values from the Terra satellite (54 - 66 µm) on all three days.

In contrast, Aqua showed $r_{\mathrm{opt}}$ between 94 to 114 µm for the same period. In all three flights, the SMART retrieval, based on the albedo measured at 1700 nm wavelength, gave smaller values than the ratio method. For 27 March, the ICON-ART simulations covered the entire flight track, since the model-based land mask of ICON-ART classified this near coastal region as land. Similar to the comparison over land (25 March), the model showed low variability (< 1 µm standard deviation) and lowest $r_{\mathrm{opt}}$ (< 100 µm) compared to all other methods. Due to the rather coarse resolution of 3.3 km in this model setup, the

small-scale variations present in the observations cannot be resolved properly.

### 5.3   Effect of surface roughness

The obvious larger variability of the retrieved snow grain size distribution over snow-covered sea ice (Fig. 6) is strongly linked to the macroscopic surface properties. The retrieval methods are based on radiative transfer theory assuming. In reality the sea ice surface structure observed during PAMARCMiP was rather complex. Ice drift in the Fram strait caused an increase

in surface roughness, which in turn led to an irregular snow distribution due to wind transport of snow. Since the spatial resolution of the satellite-based data is lower than the scale of typical roughness features, here only roughness effects on the SMART-based snow grain size retrieval are investigated.

   Four representative areas were selected from the flight on 25 March 2018 are marked in Fig. 6f, each showing a different regime of $r_{\mathrm{opt}}$. The corresponding fisheye camera images (Fig. 9) at these locations illustrate the characteristic surface type of

each area. For the smooth surface over land [1] all methods provided low and less variable $r_{\mathrm{opt}}$-values, while for regions with increased roughness [2] and not fully covered snow surface [4] the snow grain size provided more variable and and in average higher values. Area [3] represents a case, where the measurements were contaminated by low level clouds. Here $r_{\mathrm{opt}}$-values retrieved by SMART, were significantly biased toward low values, which emphasizes the need for a proper cloud mask.

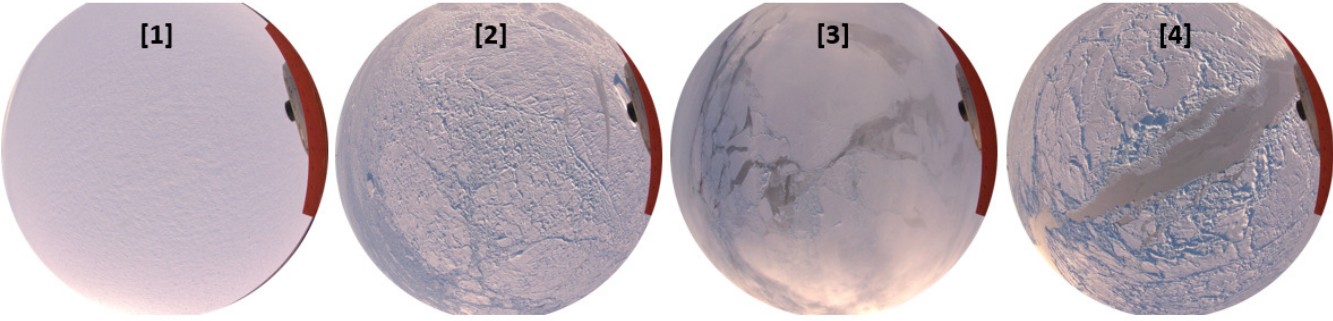

**Figure 9.** Images of surface taken by a fisheye camera at four different situations for the flight on 25 March 2018. The numbers correspond to the location numbers, which are marked in Fig. 6.





In particular, roughness features under low Sun conditions cause significant shadowed and illuminated surface elements and alter the surface reflection properties. A clear correlation between snow grain size and surface roughness data from laser scanner could not be identified in the data sets. The spatial coverage of the laser scanner data is limited to a FOV of about 60°, which not fully describes the surface conditions that affect the upward irradiance measured by SMART. Also the laser scanner

data did not cover the entire flights and therefore are limited for a statistically robust analysis.

Therefore, the surface conditions along the flight track of 25 March 2018 were classified based on fisheye camera images taken every 6 sec. Similar to Jäkel et al. (2019), the images were analyzed according to their red, green, and blue (RGB) channel values to estimate the fraction of shadowed and illuminated areas within the individual images as a marker for the roughness of the overflown surface. In contrast to the laser scanner, due to the 180° viewing angle of the fisheye lens, a direct relation

to the upward irradiance which determines the albedo and consequently the snow grain size estimate, is given. As a proxy of illuminated and shadowed areas resulting from roughness features, the ratio of the red and blue channel was calculated for each image pixel. From training images, threshold values of the ratio were defined, which characterized shadowed (ratio < 0.8) and illuminated (ratio > 1.1) pixels. The areal fractions of the shadowed and illuminated pixels ($f_{\mathrm{sh}}$, $f_{\mathrm{il}}$) were calculated with respect to the angular resolution of the image pixels (Jäkel et al., 2019). A "smooth surface" was defined when $f_{\mathrm{sh}}$ and $f_{\mathrm{il}}$

were less than 5 %, while scenes with 5 %<$f_{\mathrm{sh}}$ and $f_{\mathrm{il}}$<10 % were classified as "light rough". "Rough snow" conditions were defined for images with $f_{\mathrm{sh}}$ and $f_{\mathrm{il}}$ larger than 10 %. Further, the images indicated as "smooth surface" were subdivided into sea ice surface and land surface. As shown in Fig. 9, also regions with young ice contributions, where snow cover is greatly diminished, are included in the data set. Therefore, also a "young ice" class was introduced based on manually selecting red, green, and blue channel thresholds, derived from training samples as already applied in Jäkel et al. (2019) and Hartmann et al.

20  (2020).

The applicability of this roughness classification was tested on laser scanner data and images extracted from the downward looking video camera. Both instruments have a similar field of view in the range of 60°, allowing a direct linking of the data. The surface roughness in units of cm and the fraction of shadowed (illuminated) image pixels did show a positive correlation with a correlation coefficient of 0.68 (0.70), demonstrating that $f_{\mathrm{sh}}$ and $f_{\mathrm{il}}$ serve as representative roughness parameters for

sea ice conditions as observed during PAMARCMiP. It should be noted that for radiative effects the surface roughness as given from the laser scanner is a parameter that needs to be analyzed with respect to the orientations of the surface features and their relation to the Sun azimuth and zenith position. Therefore, the identification of shadows from the camera images is well suites to quantify the surface roughness affecting the surface radiative properties.

To estimate the impact of the surface conditions on the retrieved snow grain size from SMART measurements, the relative

contributions of the 5 individual subclasses on predefined $r_{\mathrm{opt}}$-bins (20 μm width) is shown in Fig. 10. The lowest $r_{\mathrm{opt}}$-values (70 and 90 μm bins) were only derived for observations over the smooth snow land surface. Their percentage decreased rapidly with increasing snow grain size, while the contribution of measurements over the smooth snow sea ice surface dominated the $r_{\mathrm{opt}}$-bins of 130 and 150 μm. Significant percentages of light rough and rough snow were found for $r_{\mathrm{opt}}$-bins between 110 and 230 μm indicating their variable impact on the snow grain size retrieval, where light rough snow tends to be more relevant for





smaller $r_\mathrm{opt}$-values then scenes classified as rough snow. However, highest $r_\mathrm{opt}$-values were retrieved over young and thin ice with a low snow depth, when ice absorption alters the spectral signature and distorts the relationship between surface albedo and $r_\mathrm{opt}$. It clearly demonstrates that small scale variations in snow coverage, for example, due to refrozen leads, needs to be excluded from the snow grain size evaluation. A similar study for the two flights on 26 and 27 March could not be done due to
the lack of continuous fisheye camera data.

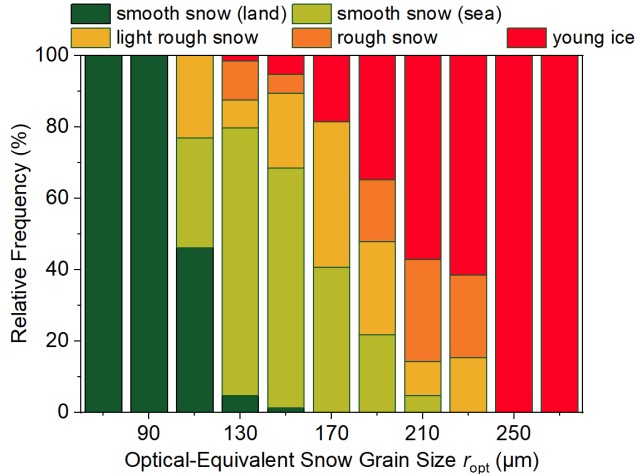

**Figure 10.** Relative frequency of $r_\mathrm{opt}$ (SMART retrieval) seperated into different surface conditions normalized to $100\,\%$ for each $r_\mathrm{opt}$-bin (25 March 2018).

## 5.4  Effect of snow particle shape

The previous MODIS and SMART based retrievals of the snow grain size were performed for a mixture of particle shapes. The sensitivity of the retrieval methods to the assumed snow particle shape was quantified for the PAMARCMiP specific
conditions with a SZA of 80° on basis of TARTES simulations. In TARTES snow albedo was calculated for different shapes such as cylinders, spheroids, cuboids, hexagonal plates with variable aspect ratios (height to the length), and fractals for $r_\mathrm{opt}$ up to 200 μm. These simulated spectral snow albedo served as input for the snow grain size retrieval, which was applied for each shape-specific TARTES simulations. The ratio of the retrieved snow grain size (using the albedo at 1700 nm wavelength and the albedo ratio $\mathcal{R}$) and the reference snow grain size (mixed shape) of the TARTES simulations is shown in Fig. 11a.
All together, for the PAMARCMiP specific conditions, the effect of the unknown snow grain particle shape may lead to uncertainties in the range of 35 % in extreme cases when using LUTs based on calculations for a mixed shape particle type, which is significantly higher than reported by Picard et al. (2009) with uncertainties of $\pm\,20\,\%$. The tendency of the deviation strongly correlates with the form factor $A$. Keeping the surface albedo constant, when absorption is enhanced due to the increase of the form factor $A$, requires a decrease of the snow grain size to compensate the absorption effect. Further we can



conclude from Fig. 11a, that there is no clear particle type (e.g., cylinder, spheroid) specific tendency of the snow grain size deviation. Rather, the particle aspect ratio may determine the tendency and magnitude of the snow grain size deviation in the same order than the particle type itself. For example, for hexagonal plates the lowest aspect ratio gave a smaller retrieved snow grain size than the mixed shape approach, while for larger aspect ratios the opposite relation was observed, which clearly is

driven by the dependence of the asymmetry parameter $g$ on the particle aspect ratio (see Fig. 7 in Neshyba et al., 2003).

It was found, that in general, the relative retrieval biases are almost similar for small and large snow grain sizes. The 1700 nm - based retrieval shows only a variability of 4 % within the studied range of $r_{\mathrm{opt}}$, while in the $\mathcal{R}$-based retrieval this spread is even half (2 %), as illustrated by the standard deviations of the $r_{\mathrm{opt}}$-averaging in Fig. 11a. Both retrieval approaches show similar results for all snow shapes. The differences for the $\mathcal{R}$-based retrieval are only slightly higher, what indicates that in

both retrievals the assumption of snow grain shape is crucial and may lead to systematic biases. However for most retrievals no a priori knowledge of the snow shape is available and the shape mixture is still the best choice. Therefore, the retrieved $r_{\mathrm{opt}}$ should be interpreted also as an shape equivalent grain size representing the snow albedo that can be calculated assuming a shape mixture.

Based on the two extreme snow shapes (hexagonal plates with an aspect ratio of 2 and cylinders with an aspect ratio of 0.25),

the retrieval algorithm was adapted to either of both by adjusting the form factor $A$. These modified retrievals were applied to the case of 25 March 2018 separating observations over land and sea ice. The statistics of the retrieved $r_{\mathrm{opt}}$ are given in 11b. Absolute mean differences of about 90 μm (1700 nm - based retrieval) and 110 μm ($\mathcal{R}$-based retrieval) over sea ice were derived, while over Greenland the mean differences decreased to 45 μm and 60 μm, respectively. This promotes the usage of the 1700 nm - based retrieval for cloudless conditions, because of its lower sensitivity to the snow grain shape than the $\mathcal{R}$-based

retrieval.

## 6  Summary and conclusions

This study compares snow grain size estimates from different observational methods and models under low Sun conditions. The analysis is based on airborne and ground-based observations during the PAMARCMiP 2018 campaign hold in the vicinity of the Villum research station, North Greenland, in early spring 2018. The applied methods to retrieve $r_{\mathrm{opt}}$ are in general all

based on optical measurements making use of the grain size dependent absorption of solar radiation by snow, but in detail depend on the specific instrument, which cover ground-based in situ (IceCube), airborne (SMART) and satellite observation (MODIS on Aqua and Terra, SLSTR on Sentinel-3). The different retrieval methods rely on the asymptotic radiative transfer theory (Kokhanovsky and Zege, 2004) applied to airborne albedo and MODIS reflectance measurements (SGSP retrieval, Zege et al., 2011), as well as a minimizing approach of measured SLSTR and pre-calculated reflectances for variable grain sizes and

shapes (XBAER retrieval). The results were compared to $r_{\mathrm{opt}}$ simulations from the ICON-ART model and a parametrization proposed by Flanner and Zender (2006). To our knowledge, these different methods have not been compared at high latitudes (low Sun conditions) before. In particular, the retrievals using albedo and reflectance measurements are subject to significant uncertainties due to the large SZA of about 80° as present during the PAMARACMiP campaign. However, conditions with

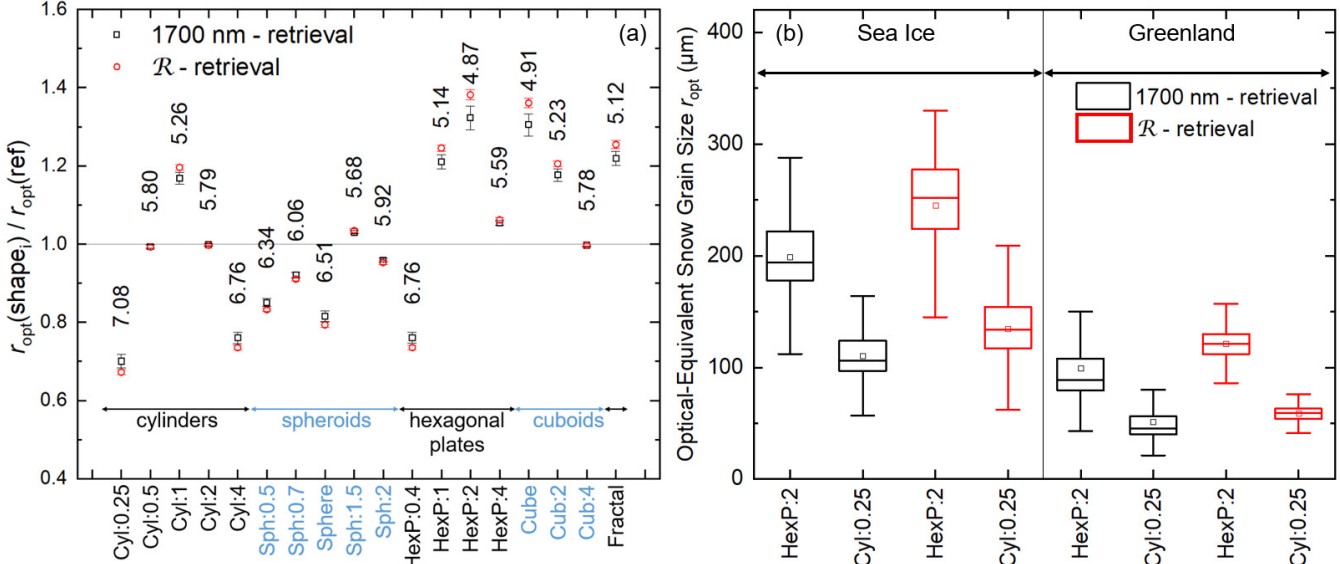

**Figure 11.** (a) Ratio of retrieved snow grain size for various particle shapes and the reference mixed shape based on LUTs from TARTES simulations for SZA = 80°. The shape-dependent form factors $A$ are given within the plot next to the $r_{opt}$-averaged values. The vertical bars indicate the standard deviation of the $r_{opt}$-averaging. The studied shapes: cylinders (Cyl), spheroids (Sph), hexagonal plates (HexP), cuboids (Cub), and fractals are selected according to the TARTES internal shape list from Libois et al. (2013). The number behind the shape abbreviation gives the aspect ratio of the particle. (b) Retrieved $r_{opt}$ from SMART measurements over sea ice and Greenland on 25 March 2018 assuming the two particle shapes hexagonal plates and cylinders.

low Sun are common in early spring in the central Arctic. Therefore, this uncertainty analysis of the different approaches is a measure to quantify measurement based uncertainties of the snow evolution, and its impact on the solar radiative fluxes by altering the surface albedo.

For the airborne snow grain size retrieval, the modified SGSP method was applied for two wavelength settings. (i) a ratio
5  method as proposed by Carlsen et al. (2017), and (ii) a single-wavelength approach at 1700 nm wavelength. An uncertainty analysis for the airborne SMART albedometer for the specific measurement conditions showed a significantly larger relative bias of the ratio method (up to 100 % for small $r_{opt}$) than for estimates using the albedo at 1700 nm (less than 25 %). The generated LUTs were calculated from the combined TARTES and libRadtran simulations accounting for the specific illumination conditions in terms of the direct-to-global fraction. Other than the intercomparison of $r_{opt}$-retrievals shown by Carlsen et al.
10  (2017) for the Antarctic Plateau, the PAMARCMiP measurement in low altitude required a full atmospheric correction, which leads to a reduction of the retrieved $r_{opt}$ by about 15 % .

Local in situ measurements over the three-week period of the PAMARCMiP campaign revealed a minor increase of $r_{opt}$ within five days after snowfall by about 15 μm, which is lower compared to measurements by Carlsen et al. (2017) on the Antarctic Plateau under a similar temperature regime (daily $r_{opt}$-increase of 5.8 μm day$^{-1}$). The $r_{opt}$-evolution modeled by





ICON-ART showed good performance for the time frame of snowfall events. In the cloudless period of the campaign, in contrast to the IceCube in situ data, the modeled $r_{opt}$ doubled its value within five days. Adjusting the growth rate factor from 0.06 $\mu m^2\,s^{-1}$ from Essery et al. (2001) to 0.012 $\mu m^2\,s^{-1}$ led to the best agreement with the in situ data. A $r_{opt}$-growth of similar magnitude as shown by the original ICON-ART simulations was derived from the parametrization after Flanner and Zender

(2006) when assuming a vertical temperature gradient in snow of 0.5 $K\,cm^{-1}$, which was close to snow pit measurements of about 0.4 $K\,cm^{-1}$. A second simulation with 0 $K\,cm^{-1}$ reproduced the measured evolution best, indicating that at these low temperatures (T < -25°C) curvature growth might have the dominant effect on the grain size evolution. Further, the relation between measured broadband snow albedo, $r_{opt}$-evolution, and illumination conditions was discussed by separating the effects of the two parameters. It was found, that the observed decrease of the broadband surface albedo by 0.07 can be predominantly

attributed to the cloud effect on the surface albedo, while the 10 $\mu m$ increase of $r_{opt}$ can only explain a broadband albedo decrease of 0.01.

Three days of cloudless conditions were selected to compare ground-based, aircraft and satellite observations of $r_{opt}$. Measurement flights over the Fram Strait performed on 25 March 2018, indicated higher and more variable $r_{opt}$-values over the sea ice ($r_{opt}$ < 300 $\mu m$) than over land ($r_{opt}$ < 100 $\mu m$), which was also deduced from the two satellite-based retrievals, XBAER

(SLSTR on Sentinel) and SGSP (MODIS on Aqua and Terra). In contrast, near coastal flights over fast ice to the North of Greenland have shown similarly low $r_{opt}$-values in the range of 100 $\mu m$. This increase of $r_{opt}$ over the sea ice was investigated by means of the higher spatial resolution of the aircraft observations with respect to surface roughness effects. Fisheye camera data were used to classify the surface roughness. We found lower $r_{opt}$-values over smooth land surfaces than over smooth sea ice, and an indication of increased $r_{opt}$ for higher fractions of surface roughness. Largest $r_{opt}$-values were derived for scenes

including young ice with a low snow depth, where the retrieval assumption of a thick snow layer is violated and systematic retrieval biases occur. However, a functional correlation of roughness fraction and the retrieved $r_{opt}$ could not be established from the available PAMARCMiP data sets.

At high latitude several satellite overpasses per day are available, such that the retrieved $r_{opt}$ of individual overpasses with different Sun geometries can be compared. Applying the same SGSP retrieval method on successive satellite observations did

not necessarily reveal the same results, as shown by a statistical analysis of data covering the flight path of the Polar 5 aircraft. The mean snow grain size deviated by up to 100 $\mu m$. Mostly lower and more variable $r_{opt}$-values for the MODIS data from the Terra satellite than retrieved from the instrument carried by the Aqua satellite. For land surface measurements near the Villum research station, snow grain size from the Terra satellite ($r_{opt}$: 54 - 66 $\mu m$) showed a better agreement to the ground-based IceCube data set ($r_{opt}$: 53 - 58 $\mu m$) than the Aqua product ($r_{opt}$: 94 - 114 $\mu m$). The difference between XBAER and SGSP

snow grain size is larger compared to the difference between SGSP and SMART due to the assumption of ice crystal shape in the retrieval. For the derivation of snow grain size over sea ice, more strict filtering to avoid pure sea ice, melt ponds, or thin snow layer above sea ice is needed in the future for all retrieval algorithm. Both SMART retrieval approaches deviated by up to 40 % from each other, but ranged between the MODIS derived extremes. The simulations of the ICON-ART model showed the smallest spatial variability and lowest snow grain size ($r_{opt}$ < 100 $\mu m$) compared to the observational methods. Small scale

variations of the surface properties were not captured by ICON-ART.



It is not the intention of this study to provide a recommendation which of these different observational methods is to be preferred. The different techniques have their own advantages and weaknesses. The large variability of satellite results, applying the same retrieval method, shows their limitation in studying the day-to-day evolution of the snow grain size under low Sun conditions. As shown here for one case of PAMARCMiP, the differences of retrieved $r_{\mathrm{opt}}$ between two overpasses is huge
compared to the typical evolution of snow grain size by snow metamorphism.

Potential retrieval uncertainties based on the airborne SMART observations were analyzed. The findings of this analysis may serve as recommendations also for satellite-based applications. We propose (i) to apply an atmospheric correction, (ii) to calculate LUTs of the blue sky albedo, instead of assuming a plane albedo, (iii) to consider roughness features and their spatial proportion by collocated laser scanner and/or imaging methods covering a similar FOV, (iv) to make use of suitable
wavelengths in the SWIR to utilize the strongest sensitivity on $r_{\mathrm{opt}}$ and lower dependence on atmospheric extinction, and (v) to use a form factor representing a mixed-type of grain shapes.

*Data availability.* The airborne measurement data are intended to be published on PANGAEA.

*Author contributions.* All authors contributed to the editing of the manuscript and to the discussion of the results. EJ and MW designed this study. EJ drafted the manuscript, performed the radiative transfer simulations, and prepared the figures. AE and AH were leading the
PAMARCMiP campaign. GB, TC, and LM contributed to the interpretation of the retrieval data. GB and MZ processed the IceCube data. LI and LM compiled the satellite retrievals. AR run the ICON-ART simulations. VH provided the laser scanner data, and MS was responsible for the instrumental preparation of SMART. KN helped to define the atmospheric conditions, and SR worked on the atmospheric correction. AH processed the sun photometer data.

*Competing interests.* The authors declare that they have no conflict of interest.

*Acknowledgements.* We gratefully acknowledge the funding by the Deutsche Forschungsgemeinschaft (DFG, German Research Foundation) – project ID 268020496 – TRR 172, within the Transregional Collaborative Research Center "ArctiC Amplification: Climate Relevant Atmospheric and SurfaCe Processes, and Feedback Mechanisms (AC)[3]". We thank Franziska Nehring from FIELAX GmbH for post-processing the meteorological aircraft data and the Institute of Environmental Physics, University of Bremen for the provision of the merged MODIS-AMSR2 sea-ice concentration data at https://seaice.uni-bremen.de/data/modis_amsr2 (last access 25 September 2018). Thanks to the IMAPP
(International MODIS/AIRS Processing Package) team at SSEC (Space Science and Engineering Center, University of Wisconsin-Madison) for providing the MODIS destriping routine. Development of IMAPP is supported by NASA under grant NNX14AK06G.





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
