# Peer review of "Comparison of optical-equivalent snow grain size estimates under Arctic low Sun conditions during PAMARCMiP 2018"

_The Cryosphere, 2021_

## Referee Comment (RC1)

**Review of "Comparison of optical-equivalent snow grain size estimates under Arctic low Sun conditions during PARMARCMiP 2018" by Jäkel et al.**

**General comment**

The authors apply different methods to retrieve the optical-equivalent snow grain size over an Arctic area (part of Greenland and adjacent sea ice) from airborne and satellite albedo observations. The dataset is extremely interesting (because of the paucity of observations in that region) and challenging (because of the high solar zenith angles and the heterogeneity of the surface). However, the applied methodology has serious deficiencies. The main problems are a misinterpretation of the role of roughness on albedo, and the fact that shadows and snow-free young ice are not excluded from the retrieval and therefore cause artefacts and biases in the results (see detailed comments to Sect. 5.3) making the comparison of grain size retrievals from the different satellite scenes meaningless. I therefore recommend the editor to not accept the paper until a major revision is done and these methodological deficiencies are solved. Also, the sections describing the applied models should be clearer, and in general the paper requires a linguistic revision, as in many sections the logic of the syntax is poor.

**Detailed comments**

p.3, line 4-5: as you clarify here that the term "snow grain size" will refer to "optical-equivalent snow grain size" throughout the paper, please be coherent and replace in all following sections (including the figures) "SSA" and "optical-equivalent snow grain size" with "snow grain size".

p.4, line 30: remove ""covered the sea ice" as it is repeated immediately after.

p.4, Section 2.1: please add a description of the broadband albedo data collected at Villum (instrument used, measurement uncertainty), which are particularly important as they will be used in the analysis.

p.5, line 2: please explain what is the vertical resolution of the DEMs generated from the laser scanner observations.

p.5, line 22: " " Recent founding revealed that the accuracy of the Ice Cube measurements is lower than this 10%, even for the reported snow density of $230\pm30$ kg m$^{-3}$. See for instance Calonne, N., B. Richter, H. Löwe, C. Cetti, J. ter Schure, A. Van Herwijnen, C. Fierz, M. Jaggi and M. Schneebeli (2020). "The RHOSSA campaign: multi-resolution monitoring of the seasonal evolution of the structure and mechanical stability of an alpine snowpack." The Cryosphere 14(6): 1829-1848. This study reported that SSA measured with the IceCube was systematically higher than values computed on tomographic images, approximately by a factor of 1.3. A definitive explanation for this disagreement has still to be found. Anyhow, as also reported in the introduction, optically-based SSA retrievals (either from albedo or from the single wavelength reflectance measurement applied in the IceCube) suffer from a ±20% uncertainty (i.e. 40% uncertainty range!) when the shape of the grains is not taken into account. Thus, in this section, I would include these arguments when presenting the total uncertainty associated to SSA measurements obtained from the IceCube.

p.6, line 28: please remove "However,"

p.7, line 1: Replace "Visual camera data of the surface taken aircraft-based" with "Aircraft-based photos of the surface"

p.7, line 2: please remove "However," as you are not introducing a contraposed sentence.

p8. lines 5-7: "The temperature profile on 26 March shows the strongest inversion of all three flights with surface temperatures of -30 C and -10 C around 880 hPa corresponding to an altitude of 1.1 km. The weakest inversion of about 5K was measured on 27 March." Please consider re-phrasing the first sentence (e.g.:…three flights, with -20 C difference between the surface temperature of -30 C and the temperature at the inversion height located around 880 hPa, corresponding to an altitude of 1.1 km"), and use the same unit for temperatures here and throughout the manuscript.

p.8, line 11: here an introductory paragraph describing the modelling strategy and the role of each of the applied models would be needed.

p. 8, line 13: "To simulate the snow surface albedo, the open-source Two-streAm Radiative TransfEr in Snow model (TARTES) was used (Libois et al., 2013)." In reality, you used several radiative transfer models, either in direct mode (to simulate spectral albedo from snow grain size) or in inverse mode (to retrieve snow grain size from measured albedo). As this section is dedicated to Tartes only, it would help a lot if you clarify for what purposes Tartes is applied (it seems to me that it was used in inverted mode to retrieve snow grain size from aircraft albedo measurements, in direct mode to calculate the albedo given as input to atmospheric radiative transfer modelling, to assess the impact of clouds on snow albedo and the impact of the assumed snow grain shape on the retrieved grain size…)

p.9-10, section 3.3 and 3.4: please clarify here that ICON-ART and Flanner and Zender scheme are applied only over land, to simulate the snow grain metamorphism in the Villum's region. I would place both models in the same section ("e.g. "modelling of the snow grain size metamorphism"?), as they will be used only for this purpose.

p. 9, line 30: "The acceleration of growth due to rain was additionally added and the reduction by snowfall was adjusted." Could you please be more specific? If this is the first time that this parameterization is applied, it has to be detailed in such a way that reader can reproduce it.

p.10, line 4: replace "temporal" with "temperature controlled" (moving it before "SSA evolution")

p.10, line 21: "Snow grain size and snow particle shape are then obtained by minimizing the surface directional reflectances at two wavelengths": I believe that this statement is wrong. Maybe you wanted to write something like "Snow grain size and snow particle shape are then obtained by minimizing the differences between modelled and observed surface directional reflectances at two wavelengths"?

p.10, section 4.2: here the applied terminology for various types of albedo is very unclear. If I interpreted correctly, I recommend using:

- "Black-sky albedo, corresponding to directional-hemispherical reflectance in case of only direct illumination" instead of "plane surface albedo, corresponding to the hemispherical reflectance and assuming only direct illumination"
- "White-sky albedo, corresponding to bi-hemispherical reflectance" instead of "spherical albedo"
- "Blue-sky albedo, corresponding to the actual albedo, i.e. a linear combination of black-sky and white-sky albedo" instead of "weighted albedo" (as it appear in Fig 2).

p.11, line 8: replace "as approximated by (Kokhanovsky, 2003)" with "as approximated by Kokhanovsky (2003)"

p.12, section 4.3.1: this section is carelessly written and extremely unclear, in both the logic and the language. Are you trying to describe two retrieval methods, one using TARTES and the other called "modified SGSP"? If so, state it, and explain the difference between the two methods. Here are some examples of unclear and inaccurate expressions (there are more, so please rewrite this section completely, with a much cleared logical structure and with precise statments):

- "the albedo ratio (R) of the SMART measurements at Lambda1 = 1280 nm and Lambda2 = 1100 nm wavelength" maybe it means "the albedo ratio (R), which is the ratio between the SMART albedo measurements at..."??

-"to minimize the retrieval uncertainty, which is affected by the measurement uncertainty of the spectral albedo" You cannot expect that the readers go to read Carlsen et al. (2017) to understand the meaning of this sentence. Please explain.

-" These aircraft measurements were performed over the Antarctic Plateau under clean atmospheric conditions" Do you mean that "Eq. 9 was applied by Carlsen et al. (2017) to retrieve snow grain size from aircraft observations over the Antarctic Plateau and, thus, at high elevation and in very dry air conditions"? High elevated and dry air, not clean... that is the point, to avoid the need of atmospheric correction.

- "…for the conditions during PAMARCMiP…" which conditions?

-"Different to Carlsen et al. (2017), in this study TARTES simulations were performed together with libRadtran calculations to generate LUTs accounting for the specific atmospheric conditions during the PAMARCMiP aircraft observations." In which way it is done differently from Carlsen et al. (2017)? Is it necessary here to know? What these LUT tables include? albedo versus snow grain size??

p. 13, line 5: "The albedo parametrization used in Sec. 4.3.1 is valid for observations at the surface". First of all the parameterizations are two, and they are not used, but "described" in Sec 4.3.1. Secondly, the statement is not true, as 1) SGSP was used for aircraft-based observations (Carlsen et al. 2017) and 2) the derived TARTES LUT tables fit to the atmospheric conditions observed during aircraft observations. I understand that you mean that, in the observed Arctic conditions, you need to apply the atmospheric correction for aircraft observations. Well then write it clearly, being rigorous and correct in your statements.

p.14, line 3-4: remove "at both wavelengths".

p.14, Fig 3: use a different line for simulated albedo at 3000m altitude and surface albedo with ropt=60micrometers. In the caption note, replace "2000" with "200".

p.14, line 14: replace "wavelength at 1700nm" with "1700 nm wavelength"

p.15, line 7: "i.e. decreasing optical depth of the snow layer." It should be "increasing" optical depth, but I would remove this part of the sentence, you have already explained the concept and this addition would not clarify anything.

p.15 Section 4.3.4: Also this section is particularly badly written, it needs to be reformulated. Especially the logic of the syntax is poor.

p.15, line 16: remove ",respectively" at the end of the sentence.

p.15, line 17: replace "form" with "from"

p.15, line 19: replace "is affected by" with "propagates from"

p.15, line 22-24: "Figure 4a compares the true snow grain size (without variation) and the uncertainty range of the retrieved snow grain size including variation in both directions (±α)." The content of this sentence is expressed quite poorly: you probably mean that "Figure 4a shows the snow grain sizes calculated from the measured albedo (using Tartes) and from R (using the modified SGSP method) as well as the associated grain size uncertainties due to the errors in the measured albedo and R."? Please rewrite. Figure 4a is quite unclear:

– The use of "R" and "1700nm" to indicated when SGPS and Tartes methods are applied to retrieve $r$ is confusing: it would be preferable to directly use Tartes and SGPS acronyms (e.g. $r_{Tartes}$ and $r_{SGSP}$)

– Do the two methods provide exactly the same $r$? If so, you must have applied some tuning, playing e.g. with the grain shape. You need to clarify this, and report how the match was achieved.

– Due to the unclarity of Sect 4.3.1, I could only guess which scheme you are applying for which wavelength (Tartes for α at 1700 nm and SGSP for R?) The logic between sections is broken, and the reader cannot really follow which scheme is applied where.

– The concept of "true optical-equivalent grain size" in x-axis is very ambiguous, please use a more correct terminology, something like "Retrieved optical-equivalent grain size from the measured albedo applying Tartes ($r_{Tartes}$) and SGSP ($r_{SGSP}$)"

– In the y axis, I would write something "Retrieved optical-equivalent grain size from the upper and lower boundaries of the albedo uncertainty interval applying Tartes and SGSP".

p.15, line 26: "…with a larger slope of the LUT (Fig. 2)." Only now the reader starts to guess what this LUT may include. Here you need to be rigorous with the syntax: you don't really mean the slope of the LUT, but the slope of the curve representing the $r_{opt}$-albedo relationship in the LUT. Right?

p.15, line 26-27: "The relation between snow grain size and surface albedo for 1700 nm wavelength shows a stronger decrease than using R." Again there is a syntax problem: please be rigorous in the logic of your sentences, here and throughout the manuscript (I will not point to all the mistakes!). Maybe here you mean that "the $r_{opt}$ decrease with the increase of albedo at 1700 nm is steeper that the $r_{opt}$ decrease with the increase of R."

p.15, line 27-28: "This leads to a higher sensitivity on the SMART measurement uncertainty applying R, and an absolute error, which is about three to five times higher than using (1700 nm) for the snow grain size retrieval." From your previous sentence, it seems to me that the opposite should be concluded. Please replace "applying R" with "applying SGSP", and "using 1700nm" with "using Tartes" (also in sentence below, in line 31): isn't it so that you are here applying two different methods, Tartes and SGSP??

p.17, line 13: "Fig 5 displays the range…" Please explain what causes this range (uncertainty range in some input parameter?)

p.18, lines 1-27: again a paragraph that is very painful to read. Please rewrite it with rigorous logic (below are just few suggestions, the whole section need to be reformulated).

p.18, line 6-9: "The sensitivity of tau (T) and kappa (T) to temperatures in the range between -20 °C and -50 °C (see Tab. 4 in Flanner and Zender, 2006), differs from the parametrization for the range between -20 °C and 0 °C, resulting in this lower fitting quality. This likely is caused by the suppression of the temperature gradient effect at -50°C, which is strongly affecting the parametrization between -20 and -50°C." Did you skip some logical steps? The meaning of this paragraph is totally obscure. Should "sensitivity" replace "parameterization"?

p.18, line 14: "Radiative transfer simulations…": using which model? TARTES?

p.18, line 16: "While period I (22 March) was characterized by a SZA of 83 °": This cannot be, the period was overcast! This means, that the effective SZA is of the order of 52-55°. You should re-calculate the albedo with the correct SZA for overcast conditions.

p.18, line 17-18: "The simulations revealed differences of 0.09 between period I (α = 0.96) and period II (α = 0.87), which are higher than observed from the pyranometer measurements (0.07)." What is the observed albedo (by pyranometers in Villum) in these two cases, and what is its uncertainty? In the best case (with high standard and freshly calibrated sensors), the measured albedo has a 1% uncertainty in overcast conditions and 5% uncertainty in clear sky (due to the low SZA), so I consider a 0.02 difference in overcast-clear albedo between model and observations insignificant. However, it seems to me that the modelled albedo is much higher than the measured albedo. Can you hypnotize an explanation for this?

p.18, line 23: "period II was re-simulated using the spectral surface albedo from TARTES, which assumed only a diffuse component (spherical albedo) due to the impact of the cloud layer from period I". Very poorly written sentence: you mean "spectral albedo in Period II was re-calculated with TARTES assuming diffuse incoming irradiance to simulate overcast conditions"?

p.18, line 30: replace "…using satellite data" with "…using MODIS and Sentinel-3 data, respectively,"

p-19, lines 2-4: "Note, that the ICON-ART simulations were restricted to areas where the model-based land mask indicated land surfaces only, and are not included in the analysis here." Please remove this sentence, it is totally out of place here, especially if you clarify from the beginning that ICON-ART and Flanner & Zender are only applied over land.

p.19, fig 6: it would very much helpful if the coastline contour of Greenland would be marked in the maps. Currently, there is black contour that is not explained, and it seems that does not correspond to the coastline (what is it?). Latitude and Longitude are inverted in x and y-labels of the plots.

p.21, line 10-11: "The best agreement to IceCube measurements (ropt = 53 µm) on this day were derived from the 16:45 UTC MODIS overpass (ropt = 54 µm)". When you add the uncertainty in the IceCube measurement (see also my comment below related to Fig. 8) you may conclude that 2-3 ropt estimations fall into the uncertainty range of the in situ observations.
p.21, line 20: replace "…leads to a large diversity, compared to the SGSP and SMART results" with "…leads to a large difference between SGSP and SMART results" (if I well understood the meaning…).

p.22, fig 8: please replace "IRQ" with "IQR" in subplot b. For the discussion, it would be important to include the uncertainty of the in situ optical effective radius measured with the IceCube (a minimum of ±20% uncertainty, but also keep in mind the systematic bias observed by Calonne et al. 2020).

p.23, line 12:"The obvious larger variability of…" Why obvious? I recommend removing "obvious"

p.23. line 13: "The retrieval methods are based on radiative transfer theory assuming." Uncomplete sentence, please correct.

p.23, line 14: "Ice drift in the Fram strait caused an increase in surface roughness" Increase? With respect to what, the rest of the Arctic?

p.23, line 20-24: please use the letters a), b) c) and d) to label the subplots of Fig 9. When referring to them, please indicate the correct labelling of the plot ("Fig 9a, point 1" instead of [1], etc.)

p.24, line 5: I recommend replacing "limited" with something like "not suitable".

p.24, line 17: "As shown in Fig. 9, also regions with young ice contributions, where snow cover is greatly diminished,…" I would specify that you are referring to Fig 9c and d, and I would replace "greatly diminished" with "thinner than in the surrounding areas".8

p.24, line 27-28: "Therefore, the identification of shadows from the camera images is well suites to quantify the surface roughness affecting the surface radiative properties." I don't see the point here: if you have the DEM generated from the laser scanner data, you are perfectly able to estimate the extension and orientation of the shadows once you know lat, lon, and time. You don't need the camera images for this.

p.24, line 33: please add "surfaces" after "Significant percentages of light rough and rough snow"

p.25, line 2: please replace "when" with "where"

p.25, line 3-4: "It clearly demonstrates that small scale variations in snow coverage, for example, due to refrozen leads, needs to be excluded from the snow grain size evaluation." That is certainly true, although the concept is badly expressed: instead of "small scale variations in snow coverage" you should use a more precise expression, such as "areas without snow or with thin snow layers that are not optically thick". However, as you have recognized this problem (artifact) in your effective snow grain size retrieval, you should correct for it, either excluding the thin ice areas from your analysis, or estimating the bias that this problem causes in your results, and, thus, present the results without this bias. Otherwise, you will include physically meaningless results.

Anyhow, the main problem of this section is a misinterpretation on the role of roughness on albedo. Although fraction of shadowed areas and surface roughness are correlated (because

shadows are caused by roughness) it does not mean that roughness and shadows cause the same effect on albedo. Roughness decreases the albedo (by few percent) because photons reach the surface features that receive most of the irradiance with lower zenith angle than the surface slopes that receive less irradiance. This effect should affect the measure albedo independently on the platform from which albedo is measured, so I don't see how it could explain the seen variability of snow grain size retrieved by the different methods and platforms. Shadows, on the contrary, do not decrease the surface albedo. They are areas that look darker than the surrounding because they receive less illumination, but their actual albedo is higher than the surrounding, because they only receive diffuse light, which is richer in the visible wavelengths. This means that, to estimate the snow albedo from satellite images, the shaded areas should be masked out, as they represent an artefact, negatively biasing the albedo. Since the fraction of shadows in a defined area depends on the time of the observations, it is not at all surprising that albedo (and, consequently, retrieved snow grain size) exhibits significant variations among the different satellite images when the shadow artifact is not corrected. This problem is particularly serious at high SZA, as the authors noticed. For instance, I suspect that the difference between the frequency distribution of retrieved snow grain size from Aqua scene at 10:35 and Terra scene at 11:50 (Fig 7) is due to the fact that shadows cover a larger area at 10:35 and therefore the uncorrected albedo is lower and the retrieved grain size is larger than at 11:50. Since the shadow artifact is not corrected, the comparison of grain size retrievals from the different satellite scenes is meaningless.

p.25, line 16-17: "the effect of the unknown snow grain particle shape may lead to uncertainties in the range of 35% in extreme cases when using LUTs based on calculations for a mixed shape particle type, which is significantly higher than reported by Picard et al. (2009) with uncertainties of ± 20 %." Your results are perfectly in line with Picard et al. (2009), as a range of 35% is inside the 40% range (±20%) of uncertainty obtained by Picard et al. (2009). Please correct.

p.26, line 6: after "…the relative  biases" please add "between the 1700nm and ratio R retrieval methods"

p.26, line 9-10: please correct the poorly written text, for example as "Both retrieval approaches show similar results for all snow shapes,  with the R-based retrieval being  only slightly higher than the 1700 nm retrievals. The assumption on the grain shape has a much more critical impact on the retrieval".

---

## Referee Comment (RC2)

Review of "Comparison of optical-equivalent snow grain size estimates under Arctic low Sun conditions during PAMARCMiP 2018"

The manuscript introduces quite interesting observations about an important problem—the processes controlling snow and ice albedo—at large solar zenith angle where radiative transfer calculations about the surface and atmosphere have difficulties. Unfortunately, the analyses are confusing enough that the essential messages and uncertainties are obscured. All the retrievals of grain size and shape are based on some sort of radiative transfer calculation, but the algorithms are different for IceCube, SMART, MODIS, and SLSTR. There were apparently no physical observations recorded for the grains, so retrievals about shapes are difficult to interpret. Moreover, if grain shape is important, would not the grains' orientation also be important? Whereas ice crystals in cirrus clouds have a preferred orientation, observations in snow on the ground seem less conclusive. Once snow has begun to sinter, the original shapes of the flakes during snowfall are mostly obscured.

Especially at these solar illumination conditions, macroscale roughness affects the angular distribution of the reflectance, even when the roughness elements are only a few centimeters tall. Thus the exquisite detail of the calculations, which assume a smooth surface, do not account for the roughness. I appreciate the measurements in the images that show such variability, but the main message is that the details of the retrievals might be irrelevant, especially about shape.

The retrievals from satellite imagery gloss over some important variability. The pixels sizes for MODIS on Terra and Aqua and for SLSTR on Sentinel-3 are large. To what extent does variability *within* the large MODIS or SLSTR pixel affect the results? Moreover, the MODIS and SLSTR measurements on different days have varying view angles. Whereas the different view angles measure different samples of the BRDF and thereby cause variability (Diner et al., 2005), a bigger effect likely arises from the varying pixel size, a factor as large as 10x between nadir and the edge of the scan for a nadir scan angle of 55° combined with Earth curvature (Dozier et al., 2008).

Figures 6 through 8 are truly interesting, but hard to interpret. From the discussion in the text, it would be useful to indicate the coastline in Figure 6 and thus the boundary between snow on land and snow on sea ice.

Therefore, I agree with the other referee's assessment and comments. This manuscript needs substantial reanalysis and revision before it should be published as a peer-reviewed contribution.

Detailed comments on specific text:

Page 2, Lines 28-31. The statement is incorrect: "Microscopic differences in water vapour pressure at saturation due to variable curvatures for a single snow grain cause water vapour diffusion from convex to concave parts of the snow grain. The corresponding deposition (at concave surfaces) and sublimation (at convex surfaces) changes the particle shape to more rounded particles, which have a larger grain size than initially (Colbeck, 1982; Cabanes et al., 2002)."
Calculation of this effect using the Kelvin equation shows that concavity or convexity is significant only for minute radii of curvature (<1µm). The process that causes the change in the grain shape in near equilibrium conditions (low temperature gradiant) is grain-boundary diffusion. The dihedral shape of the boundaries between sintered grains is inconsistent with the vapour-diffusion explanation, as Colbeck (1998) clarifies. The Flanner-Zender (2006) model might give reasonable results, but perhaps for the wrong reason.

Page 3, Line 17. The statement is incorrect: "All common retrieval methods rely on the same asymptotic radiative transfer (ART) approach (Kokhanovsky and Zege, 2004) …" Approaches that retrieve grain size along with fractional snow-covered area from multispectral sensors MODIS and Landsat (Bair et al., 2020; Painter et al., 2009), and the new USGS product (Landsat Fractional Snow Covered Area (usgs.gov)), use Mie scattering and the radiative transfer equation, with different assumptions about grain shape. I am not sure the distinction matters, because the different approaches give similar results, but one should be careful about any sentence that has the word "all."

Page 8, Line 15. Rather than Stamnes et al. (1988), it is probably better to cite the original delta-Eddington paper (Joseph et al., 1976), which Wiscombe and Warren (1980) applied to model snow's spectral albedo.

Page 8, Lines 28-30. For a smooth surface, we would expect that snow albedo would increase with SZA. For a rough surface, would greater shadowing at larger SZA compensate in the opposite direction?

Page 9, Line 30 and other places. The text is unclear about the model based on age and temperature to estimate change in grain size. Do you use Essery et al. (2001) or Flander & Zender (2006)? Or are the models combined? Moreover, the Essery et al. (2001) reference has no information about availability in the bibliography, like a DOI.

Page 10, Lines 21-22. Perhaps change this sentence to, "Snow grain size and snow particle shape are then obtained by minimizing the differences between theoretical simulations and SLSTR observations of surface directional reflectances at two wavelengths (0.55 µm and 1.6 µm)." (The other referee also suggests a revision of this sentence.)

Figure 3. The caption and the figure seem inconsistent. There is no "thick solid red line" in the figure itself, and the lines for "$\alpha_{3000m}$" and "60 µm (reference)" appear identical.

Pages 10-13, generally. For surface values, grain size is retrieved by a delta-Eddington approximation of the radiative transfer equation. But for the remotely sensed retrievals, the XBAER approach, which incorporates and atmospheric adjustment also, is used for the SLSTR data; the Zege et al. (2011) method is used for the MODIS data; and Carlsen et al. (2017) is used for the airborne data. The different methods treat the diffuse fraction of the irradiance differently, along with different methods of solving the radiative transfer equation. To what extent are the retrievals of grain size/shape and light-absorbing impurities affected simply by the different approaches? Figure 2 seems to address this question, but indirectly. Moreover, none of these models of snow reflectance incorporate macroscale surface roughness.

Page 20, Line 29. The word "data" is the plural of "datum" so "data was" should instead be "data were".

Page 21, Line 5. A retrieved grain size of 12 µm is exceptionally tiny, or does the 12 µm refer to some difference? The sentence is unclear.

The other referee had more comments on the later pages. I agree with them.

References cited in the review

Bair, E. H., Stillinger, T., and Dozier, J.: Snow Property Inversion from Remote Sensing (SPIReS): A generalized multispectral unmixing approach with examples from MODIS and Landsat 8 OLI, IEEE Transactions on Geoscience and Remote Sensing, https://doi.org/10.1109/TGRS.2020.3040328, 2020.

Colbeck, S. C.: Sintering in a dry snow cover, Journal of Applied Physics, 84, 4585-4589, https://doi.org/10.1063/1.368684, 1998.

Diner, D. J., Braswell, B. H., Davies, R., Gobron, N., Hu, J., Jin, Y., Kahn, R. A., Knyazikhin, Y., Loeb, N., Muller, J.-P., Nolin, A. W., Pinty, B., Schaaf, C. B., Seiz, G., and Stroeve, J.: The value of multiangle measurements for retrieving structurally and radiatively consistent properties of clouds, aerosols, and surfaces, Remote Sensing of Environment, 97, 495-518, https://doi.org/10.1016/j.rse.2005.06.006, 2005.

Dozier, J., Painter, T. H., Rittger, K., and Frew, J. E.: Time-space continuity of daily maps of fractional snow cover and albedo from MODIS, Advances in Water Resources, 31, 1515-1526, https://doi.org/10.1016/j.advwatres.2008.08.011, 2008.

Joseph, J. H., Wiscombe, W. J., and Weinman, J. A.: The delta-Eddington approximation for radiative flux transfer, Journal of the Atmospheric Sciences, 33, 2452-2459, https://doi.org/10.1175/1520-0469(1976)033<2452:TDEAFR>2.0.CO;2, 1976.

Painter, T. H., Rittger, K., McKenzie, C., Slaughter, P., Davis, R. E., and Dozier, J.: Retrieval of subpixel snow-covered area, grain size, and albedo from MODIS, Remote Sensing of Environment, 113, 868-879, https://doi.org/10.1016/j.rse.2009.01.001, 2009.

Wiscombe, W. J., and Warren, S. G.: A model for the spectral albedo of snow, I, Pure snow, Journal of the Atmospheric Sciences, 37, 2712-2733, https://doi.org/10.1175/1520-0469(1980)037<2712:AMFTSA>2.0.CO;2, 1980.

---

## Referee Comment (RC3)

Review of "Comparison of optical -equivalent snow grain size estimates under Arctic low Sun conditions during PAMARCMiP 2018", by Jäkel et al.

The manuscript tries to address the issue of snow grain size retrievals in the Arctic under low sun conditions. The topic is important and timely, considering the influence of snow on albedo and the heat budget in the Arctic, and the scarcity of data in the region.

There are several shortcomings in the manuscript and I recommend a major revision. I will summarize the major points below and then give some detailed comments.

1) I found the manuscript hard to follow and I had to reread most sections several times. The main issue is that this study uses several remote sensing products together with various radiative transfer models to estimate albedo and to infer optically relevant snow grain size. On top of that this study also uses models to estimate grain growth (or changes in the SSA/optically relevant snow grain size) as well as models that simulate changes in albedo. Then there is also the ECMWF and the ICON snow model. This is all extremely confusing and it is not clear to me how a grain growth estimate or albedo evolution "model" fits into this study. One way to improve the manuscript would be to restructure the paper and to separately summarize what is observed, what is modelled using the observations (grain size and albedo) and what is modelled using different inputs (which is the albedo evolution and SSA evolution). And then summarize how the different observations/models are combined and address what they can or cannot deliver. An overview figure or even table might help here.

2) Another problem with the manuscript is the different resolutions of the various retrieval methods. The resolution of the retrieval from a satellite borne measurements would be different from that of an airborne measurement, in that the spatial variability within each retrieved pixel is different, which will have an effect on the results. This should be discussed.

3) The manuscript refers to IceCube measurements as a ground-based reference, but fails to acknowledge that IceCube is not a direct measurement of the SSA, but has similar issues are other remote sensing methods. It should discuss and include the recent work by Calonne at al., which discusses the uncertainties of IceCube and compares the results to direct measurements of SSA from micro-computed-topography. The uncertainties are certainly larger than the here reported 10%.

4) Suppressing the vertical temperature gradient in snow over sea ice is not really an option in March. Considering that the air temperatures are around -30C and the sea ice around -2C, the temperature gradients should not be assumed non-existent. The result may fit better with the retrievals, but it would be for the wrong reasons. This should be addressed.

Detailed comments:

P1, L4: what's "partly rough"? it should be defined.

P1, L15ff: it should be mentioned here that the large spread of MODIS retrievals may have something to do with the time of day and the shadows that are present at different times of day.

P2, L28: The variable curvature doesn't necessarily refer to convex or concave shapes, but can be applied to the same "shape" with varying grain size (and therefore curvature). The effect of "equilibrium metamorphism" is also quite small. The so-called equilibrium metamorphism is caused by grain-boundary diffusion and not by vapour diffusion, as demonstrated for example by Kämpfer and Schneebeli 2009 (Observation of isothermal metamorphism of new snow and interpretation as a sintering process).

Furthermore, rounded grains and grain growth are not necessary linked to equilibrium metamorphism, but have been observed under sinusoidal temperature gradients (see Pinzer and Schneebeli 2009, Snow metamorphism under alternating temperature gradients: morphology and recrystallization in surface snow).

P3, L4-5: It's a bit confusing to call the "optical-equivalent grain size radius" grain size. It is not clear later in the manuscript to which grain size the manuscript refers. Maybe for consistency and clarity, you could just refer to SSA.

P3, L20: While I agree that the macroscopic surface roughness is relevant, I don't think that the roughness in this manuscript is the macroscopic roughness. Figure 1 shows really high values of surface roughness that is quite large. This is not surprising, considering the horizontal resolution is 1 m (p5, L2). The colour bar on Figure one make it impossible to distinguish between roughnesses that are orders or magnitude in difference. Maybe consider using a logarithmic scale as well as a discrete instead of a continuous colorbar?

P3, L20: What about grain orientation and the asymmetry factor?

P4, L10-20: This should be rewritten, because it is not clear. The order of these sections does not make much sense, the way this is written.

P4, L25: Why is a 2012 paper cited for a 2018 campaign?

P5, L12ff: see also one of the major comment about the accuracy of IceCube. This should be discussed here.

P6, L18: There is a typo here. The MODIS wavelengths are in micrometer not meter. Also the lower value is 0.405 and not 0.4 microns (for consitentcy, because the higher value is given in three digits after the comma).

P7, L4: How is this surface roughness defined? Is it the RMS height or the correlation length? Or some combination of both? This should be clarified. Also see comment for p3, L20 on how to improve the figure.

P7, L16: remove "rather".

P8, L25: So the asymmetry factor and the absorption parameter were constant for all calculations in the radiative transfer model? This is not clear from the text.

P9, L14-15: What are the default parameters?

P9, L30: I can not access the Essery 2001 reference. Can you reference something with a doi? Also ageing for albedo may work for warmer conditions, but it may not be representative for estimating albedo changes in -30C conditions. But as said, I don't have access to the reference.

P10, L23: Sensitivity study instead of sensitive study.

P12, L9: I thought TARTES was used to calculate snow surface albedo from SSA (and other snow properties). So you also use it to estimate the grain size? Can you please explain? Do you use it to calculate albedo from IceCube measurements, and then to calculate grain size from albedo retrievals?

P14, Figure 3: It would be interesting to see the signal to noise ratio at 1700 nm. Did you find any issues with this?

P15, L7: How was snow density measured? This was not mentioned before.

P15, L14: Considering the large uncertainty of IceCube, I think 5 microns in difference is extremely small. Maybe the uncertainty should be added here.

P16, L3: Are the "ground-based measurement" the IceCube measurements? Are the snow layers that are discussed here used anywhere else in this study?

P16, L16: Do "both data sets" refer to line A and line B?

P16, L22: The Essery 2001 reference that is not available is listen previously for the albedo "ageing" as well as here for the grain growth. Considering that it is important for this publication, it really should be accessible.

P17, L10-15: As mentioned above, assuming no vertical temperature gradient is not an option over sea ice in March. It really does not matter whether this model fits the observations, if it's for the wrong reasons. This needs to be fixed.

P18, L16: The solar Zenith angle should be adjusted for clouds. It certainly is not 83 degrees under cloudy conditions. It would be significantly lower. It needs fixing.

P21, Figure 7: I really like this figure. It should be given a bit more context in the text.

P21, L10-11: The agreement should be considered together with uncertainties of IceCube.

Figure 8: The differences certainly have something to do with the errors due to shadows. I am not sure how useful this analysis is if the shadows are not removed.

Figure 9: What's the resolution of figure 9? And why do you need this analysis if you have the TLS data? It's not entirely clear?

P23, L12: Unfinished sentence.

P25, L1-5: The small variations in in snow coverage do need to be excluded, otherwise I am not sure if the results from this study are valid. Same for the shadows (P29, L1-5), they should be excluded when estimating the albedo.

P28, L7: As mentioned above, it is very unlikely that the temperature gradient is zero. And even if it were, the curvature effect would not be driving the metamorphism.

P29, L2: After such a large study, the strengths and weaknesses of different methods should be summarized.